# Potential Role of Biochar and Silicon in Improving Physio-Biochemical and Yield Characteristics of Borage Plants under Different Irrigation Regimes

**DOI:** 10.3390/plants12081605

**Published:** 2023-04-10

**Authors:** Saad Farouk, Arwa Abdulkreem AL-Huqail, Seham M. A. El-Gamal

**Affiliations:** 1Agricultural Botany Department, Faculty of Agriculture, Mansoura University, Mansoura 35516, Egypt; gadalla@mans.edu.eg; 2Department of Biology, College of Science, Princess Nourah Bint Abdulrahman University, P.O. Box 84428, Riyadh 11671, Saudi Arabia; 3Medicinal and Aromatic Plants Research Department, Horticulture Research Institute, Agricultural Research Center, Giza 12619, Egypt; s_elgamal99@yahoo.com

**Keywords:** antioxidant, biochar, borage, drought, chlorophyll assimilation, silicon, water status

## Abstract

Silicon (Si) and biochar (Bc) are key signaling conditioners that improve plant metabolic processes and promote drought tolerance. However, the specific role of their integrative application under water restrictions on economical plants is not yet well understood. Two field experiments throughout 2018/2019 and 2019/2020 were conducted to examine the physio-biochemical modifications and yield attributes of borage plants mediated by Bc (9.52 tons ha^−1^) and/or Si (300 mg L^−1^) under different irrigation regimes (100, 75, and 50% of crop evapotranspiration). Catalase (CAT) and peroxidase (POD) activity; relative water content, water, and osmotic potential; leaf area per plant and yield attributes; and chlorophyll (Chl) content, Chl_a_/chlorophyllide_a_ (Chlid_a_), and Chl_b_/Chlid_b_ were considerably reduced within the drought condition. On the other hand, oxidative biomarkers, as well as organic and antioxidant solutes, were increased under drought, associated with membrane dysfunction, superoxide dismutase (SOD) activation, and osmotic adjustment (OA) capacity as well as a hyperaccumulation of porphyrin intermediates. Supplementation of Bc and Si lessens the detrimental impacts of drought on several plant metabolic processes associated with increasing leaf area and yield attributes. Their application under normal or drought conditions significantly elicited the accumulation of organic and antioxidant solutes as well as the activation of antioxidant enzymes, followed by lessening the formation of free radical oxygen and mitigating oxidative injuries. Moreover, their application maintained water status and OA capacity. Si and/or Bc treatment reduced protoporphyrin, magnesium-protoporphyrin, and protochlorophyllide while increasing Chl_a_ and Chl_b_ assimilation and boosting the ratio of Chl_a_/Chlid_a_ and Chl_b_/Chlid_b_, resulting in a rise in leaf area per plant and yield components following these modifications. These findings highlight the significance of Si and/or Bc as (a) stress-signaling molecule(s) in regulating defensive systems in drought-affected borage plants by boosting antioxidant aptitude, regulating water status, and accelerating chlorophyll assimilation, thus leading to increasing leaf area and productivity.

## 1. Introduction

Climate change, water scarcity, urbanization, and population growth have undoubtedly impeded agricultural production in the twenty-first century [1,2]. Within the next 100 years, it is predicted that two-thirds of the world’s population will experience longer and more frequent drought, which will have an immediate impact on both groundwater and surface water [3]. Thus, scientists are constantly attempting to develop ongoing initiatives for reducing such dramatic repercussions [2,4].

Drought represents the biggest global obstacle to crop sustainable production [2,5]. Drought affects about 25% of the world’s agricultural fields, and by 2090 it is anticipated to affect 30–40% [6]. The yield of several crops has decreased by 17–70% as a result of the drastic effects of drought on morpho-biochemical and molecular issues [2,7]. In addition to interfering with plant development, it reduces productivity by lowering plant biomass, relative water content (RWC), chlorophyll level, water potential, cell division, and net photosynthetic capacity [8,9]. Additionally, drought stress evokes the assembly of reactive oxygen species (ROS), which severely oxidize proteins, DNA, and lipids [10,11]. Hence, ROS mitigation is the supreme imperative protection strategy for coping with environmental anxiety [12,13].

Given that the agriculture sector uses 85% of all water resources and that the common on-farm irrigation systems are insufficient and have inadequate irrigation requirements, agricultural extension in Egypt needs a significant volume of irrigation water that is currently insufficient to meet conventional needs. Recent increases in competition for limited water resources have encouraged the profitable innovative strategy for maximizing crop productivity and water use efficiency (WUE). Deficit irrigation should be recognized as incredibly useful, simple, and essential for improving WUE [14,15]. The development of drought-tolerant genotypes (using traditional or modern breeding techniques) requires excessive effort, a long time, and a lot of money, which are only available for short-term solutions [16]. Conversely, there is an excessive attitude in which promoting substances are utilized to combat drought stress. Accordingly, the spraying of nutrients or soil amendments represents more rational approaches [2,17], but there has not been much work focused on biochar (Bc) and/or silicon (Si) in arid and semi-arid areas.

The addition of organic material is a familiar and traditional approach for improving soil fertility, which therefore motivates plant development, stress resistance, and nutrient use efficiency. Recent reports have shown that using carbon-rich resources, i.e., Bc, lignite, and compost, can lessen the deleterious injuries of environmental stresses [11,17]. Just 7 million tons of Egypt’s agricultural waste (30–35 million tons) are used as animal feed, while 4 million tons are used as organic manure [18]. So, using the Bc from these agro-wastes is the best technique to manage agricultural residues. Currently, Bc (black gold) is being used increasingly in agricultural sectors, produced by the thermal pyrolysis of organic wastes in the absence of oxygen or under oxygen-limited circumstances [19]. The current marketplace price of Bc accessible by agricultural applications ranges from USD 300 to 500 per Mg^−1^ in the US [20], and in Europe, consistent with Schmidt and Shackley [21], the price of Bc was found to be as low as USD 200 per Mg^−1^. Several investigations reported that Bc application under stress conditions improved plant establishment and nutrient availability [11,17]. Due to its functions in reducing the effects of climate change, managing waste, improving soil fertility, sequestering carbon, assembling bio-energy, immobilizing contaminants, and enhancing crop growth and production, Bc has received a lot of attention [17,22]. Aside from its encouraging impact in boosting greenhouse gas sequestration, Bc, when used to amend soil, can improve soil physio-chemical properties, surface areas, water retention, and soil cation exchange capacities [23,24]. Several studies have stated that Bc supplementation enhanced photosynthesis in drought-affected crops [22,25]. Akhtar et al. [26] established that Bc considerably increased chlorophyll accumulation, stomatal conductance (Gs), photosynthetic rate, WUE, and RWC, and improved the stomatal density of drought-affected tomato plants. Additionally, Bc supplementation improved the RWC, transpiration rate, and osmotic potential of drought-affected plants relative to untreated plants [11,22].

Next to oxygen, Si represents the second utmost abundant element within the Earth’s crust and is a prime nutrient for several crops [27]. Silicon’s role in plant metabolic pathways has recently drawn increased attention. As a result, Si may play a role in biochemical processes and/or structural activities in drought-affected sweet basil [15]. Moreover, it stimulates plant growth, development, and productivity [2,8] and promotes enzymatic activity [2,22] and gene expression [28]. It has been reported that Si has been shown to have the benefit of moderating environmental stresses including drought [2,8,13] via osmolytes accumulation and the modulation of antioxidant enzymes activity [2,15]. Conferring to earlier findings, Si spraying can restore the aptitude for ROS mitigation by regulating the activity of antioxidant enzymes [2,22]. Additionally, Si’s primary mechanical-physical roles can improve crops’ ability to withstand a variety of stresses [29]. This Si-mediated encouraging effect might be attributable to raised leaf rigidity [2,15,22], thus delaying leaf senescence and raising photosynthetic pigment assimilation [22,30]; increased ribulose-bisphosphate carboxylase activity [31]; and/or smoothing light diffusion, which subsequently leads to a greater rate of photosynthesis [22,30]. 

Borage (*Borago officinalis* L.), a vegetable or oil-seed crop, has long been used in traditional medicine to relieve respiratory problems, coughing, swelling, and inflammation [32,33]. Borage oil contains considerable levels of α-linolenic acid, a critical fatty acid necessary for the nourishment of all animals and humans and an essential mediator of vital compounds in the body, including prostaglandin E1 and its derivatives [32,33]. Additionally, due to their low-cost, non-corrosive, and pollution-free nature, vital fatty acids have a significant role in preventing cancer, cardiovascular disease, and infectious disorders [34,35]. 

Nevertheless, little is implicit about the functions of Si and Bc in mitigating drought-related damage in borage plants. Consequently, the purposes of the current investigation were to: (1) examine the impact of Si and Bc on the yield, water status, chlorophyll and solute accumulation, and antioxidant capacity of borage plants; (2) assess whether Si and Bc are extra effective in lessening drought in borage; and (3) elucidate potential strategies by which Si and Bc lighten drought injuries in borage plants.

## 2. Results

### 2.1. Oxidative Destruction Biomarkers

To ascertain the efficiency of Bc and/or Si and their combinations in minimizing drought-evoked oxidative damage, hydrogen peroxide (H_2_O_2_), malondialdehyde (MDA), protein carbonyl group (PCG) assimilation, and electrolyte leakage (EL%) were examined (Table 1). Over the control plants, drought stress accelerated the buildup of H_2_O_2_, MDA, PCG, and plasma membrane dysfunction. Within severe drought, the concentrations of H_2_O_2_, MDA, PCG, and EL%, increased by 98.78 and 96.14%, 51.28 and 66.89%, 46.01 and 51.70%, as well as by 12.42 and 14.70% in the first and second years, correspondingly, compared to all control plants. In comparison to control plants, the supplementation of Si and/or Bc dramatically decreased H_2_O_2_, MDA, and PCG, as well as EL%. In moderate or severe drought, the application of Si and/or Bc lowered these values relative to the untreated plants under corresponding irrigation regimes. Employing a combined Si + Bc treatment produced the maximum reduction in oxidative biomarkers, resulting in decreased H_2_O_2_ (28.80, and 29.90%), MDA (24.41, and 26.53%), PCG (46.98, and 48.52%), and EL% (8.88, and 10.29%) in both years, accordingly, compared to non-treated severe drought-affected plants (Table 1). These findings demonstrate that Si + Bc supplementation might lessen drought-induced ROS assembly and the consequent oxidative damage in borage plants.

### 2.2. Antioxidant Enzyme Activities

Compared to untreated well-watered borage plants, Si and/or Bc supplementation enhanced antioxidant enzyme activity. The maximum activities of catalase (CAT) and peroxidase (POD), along with the lowest activity of superoxide dismutase (SOD), were obtained using the Si + Bc treatment compared to untreated plants. It is manifest that drought accelerates SOD activity, whereas the activities of CAT and POD were considerably decreased compared to their relevant control plants (Table 2). Severe drought was the most harmful compared to moderate drought or well-watered plants. The supplementation of Si and/or Bc in moderate or severe drought-affected plants further enhanced CAT and POD activities and lowered SOD activity compared to drought-affected plants only (Table 2). Compared to untreated plants under moderate or severe drought stress, the application of Si + Bc is the most efficient method for enhancing antioxidant enzyme activity.

### 2.3. Antioxidant Solutes

The addition of Si and/or Bc boosted antioxidant solute concentrations in borage plants when compared to untreated well-watered plants. Using Si + Bc above untreated plants yielded the highest concentrations of ascorbic acid (17.0 and 17.17 mg g^−1^ FW), total soluble phenolic compounds (20.48 and 20.83 mg gallic acid g^−1^ FW), flavonoid (3.522 and 3.730 g quercetin g^−1^ FW), and total anthocyanin (5.368 and 5.522 mg 100 g^−1^ FW) in the corresponding years (Table 3). It is shown that in comparison to their relevant control without Si and/or Bc supplementation, drought results in a hyperaccumulation of antioxidant solutes (Table 3). As for the interaction impacts, the data in Table 3 indicate that adding Bc and/or Si to such water regimes significantly boosted antioxidant solutes compared to untreated plants within moderate or severe drought conditions. The highest values of phenols, flavonoid, anthocyanin, and ascorbic were detected with the supplementation of Bc + Si under severe drought, which increased them by 216.1%, 192.5%, 212.8%, and 177.2% in the first year and by 214.7%, 181.0%, 204.4% and 174.6% in the second year, respectively (Table 3).

### 2.4. Organic Solutes

Considerable concentrations of the total free amino acid (TAA), glycinebetaine (GlyBet), and total soluble carbohydrates (TSC) were observed under drought and Si and/or Bc treatments (Table 4) compared with well-watered plants. Severe drought produced 105.15 and 102.82%; 48.29 and 47.93%; as well as 110.69 and 107.06% greater TAA, GlyBet, and TSC levels compared to well-watered plants in both years, correspondingly. Osmolyte accumulation was raised by the supplementation of Si and/or Bc relative to untreated plants. The utmost effective treatment was Bc + Si, compared to each one alone or untreated plants. As regards the interactive impacts, the application of Si and/or Bc substantially evoked TAA, GlyBet, and TSC accretion in borage under drought and well-watered conditions in relation to the plants with no modulator usage (Table 4). The supreme concentration of organic solutes was documented with Si + Bc application along with severe water deficit, which increased TAA, GlyBet, and TSC by 189, 104, and 200% in the first year and by 183, 112, and 126% in the second year.

### 2.5. Leaf Water Status

The results in Table 5 show that there are considerable modifications in the water potential (Ψw), osmotic potential (Ψs), turgor potential (Ψp), and RWC amongst well-watered and water-deficient plants, along with applications of Si and/or Bc (Table 5). Plant water status, including RWC%, Ψw, Ψs, and Ψp, dropped dramatically under drought stress. Concomitant to well-watered plants in both years, severe drought was the most effective in this regard, with decreasing RWC (15.66 and 16.69%), Ψw (61.90 and 67.02%), Ψs (70.90 and 68.08%), and Ψp (83.22 and 90.06%) in both years. Instead, the drought dramatically enhanced the osmotic adjustment (OA) capacity, which improved water absorption. Compared to moderately or well-watered plants, severe drought was linked to the highest OA. The application of Si and/or Bc significantly enhanced RWC% and produced a sharp fall in Ψw, Ψs, and Ψp, which stimulates the establishment of recovery plants and increases OA capacity. Typically, Si + Bc was the most effective treatment in both seasons (Table 5). Concerning the impact of the interactions, Si and/or Bc application (particularly Si + Bc) considerably boosted RWC while lowering Ψw and Ψs in comparison to plants that had not been treated. With Si and/or Bc, the OA significantly improved, which can be linked to the leaf’s ability to maintain Ψp. In both well-watered and drought conditions, the application of Si + Bc was typically more effective in raising leaf Ψp than either material alone (Table 5).

### 2.6. Chlorophyll Concentration and Assimilation

Table 6 shows that chlorophyll level and biosynthesis were significantly adjusted by drought and Si and/or Bc supplementation. Compared to well-watered plants, drought stress dramatically decreased chlorophyll (Chl_a_) and Chl_b_ concentrations as well as the ratios of Chl_a_/chlorophyllide_a_ (Chlide_a_) and Chl_b_/Chlide_b_ in both seasons. A substantial drop was seen during a severe drought. Alternatively, Si and/or Bc application considerably raised Chl_a_, Chl_a_/Chlide_a_, and Chl_b_/Chlid_b_, with the exception of the ratio of Chl_a_/Chlide_a_ and Chl_b_/Chlide_b_ in the first season, which non-significantly increased in comparison to untreated plants. In terms of the interaction effect, the application of Bc and/or Si within irrigation regimes considerably (*p* ≤ 0.05) increased Chl_a_, Chl_b_, Chl_a_/Chlide_a_, and Chl_b_/Chlide_b_ compared to non-treated plants within irrigation regimes.

Drought considerably raised porphyrin intermediates, i.e., protoporphyrin (Proto), magnesium protoporphyrin (Mg-Proto), and protochlorophyllide (Pchlide), according to the data reported in Table 7. The highest values were found under severe drought, which raised Proto (1.55 and 7.54%), Mg-Proto (11.74 and 18.66%), and Pchlide (44.15 and 45.11%) in both years, respectively. In both years, adding Bc to the soil and/or applying Si (Table 7) significantly reduced Proto, Mg-Proto, and Pchlide levels compared to non-treated plants. In the first and second seasons, treatment with Bc + Si substantially decreased Proto by 9.29 and 9.93%, Mg-Proto by 5.96 and 11.27%, and Pchlide by 25.76 and 24.01%. Furthermore, the same table reveals that the Bc and/or Si application significantly nullified the drastic effects of moderate and severe drought compared to untreated plants within irrigation regimes. The most effective treatment for the mitigation of the harmful effects of drought on porphyrin intermediates was Bc + Si treatment (Table 7).

### 2.7. Leaf Area

The leaf area of the borage plants decreased in both seasons due to water stress. As compared with well-watered plants, the lowest values (365.0 and 387.4 cm^2^ plant^−1^) were seen under severe drought (Figure 1a). In comparison to untreated plants, soil amendment with Bc and/or foliar spray of Si dramatically enhanced plant leaf area^−1^. The largest leaf area was seen in plants that received Bc + Si supplements, which increased it by 113.9% and 111.8%, respectively, compared to untreated plants (Figure 1b). Regarding the interaction, the results in Figure 1c show that the use of Bc and/or Si significantly lessens the drastic effect of irrigation regimes on borage plant leaf area. The supreme effective treatment was Bc + Si, which increased leaf area per plant compared to untreated plants in this irrigation regime by 152.5% and 145.5% under mild drought, and by 181.5% and 176.1% in severe drought within two years, respectively.

### 2.8. Yield Characteristics

The variance of the yield trials under irrigation regimes with or without Bc and/or Si is displayed in Table 8. A progressing water deficit triggered a substantial (*p* ≤ 0.05) decline in the yield characteristics of borage plants. Relative to well-watered plants, severe drought conditions resulted in significant reductions in plant seed yield^−1^ (26.84 and 29.53%), seed index (34.86 and 24.77%), oil % (21.78 and 21.66%), and plant oil yield^−1^ (42.71 and 44.91%) in both experimental years, respectively (Table 8).

In comparison to non-treated plants, Bc amendment and/or Si foliar treatment, and their interactions, all significantly (*p* ≤ 0.05) improved yield attributes. The highest values (Table 8) were found in Si + Bc-treated plants, which outperformed untreated control plants in both years in terms of plant seed yield^−1^ (25.87 and 31.91%), seed index (35.23 and 29.05%), oil % (23.59 and 24.69%), and plant oil yield^−1^ (55.43 and 63.63%).

Regarding interactions, the data in Table 8 show that applying Si and/or Bc considerably improved all yield trials compared to untreated plants under such drought regimes. Regarding untreated plants within irrigation regimes, Bc + Si supplementation considerably enhanced plant seed yield^−1^ (23.13 and 32.22%), seed index (23.43 and 61.59%), oil % (24.94 and 20.19%), and plant oil yield^−1^ (53.95 and 58.85%) in the first season. On the other hand, the same treatment boosted plant oil production^−1^ (64.11 and 50.10%) and plant oil yield^−1^ (31.39 and 28.61%) in the second season compared to untreated plants in moderate and severe drought, respectively.

## 3. Discussion

Environmental perturbation, including drought-evoked photo-oxidative stress as a result of low cyclic electron flow, seriously impacts bio-membrane permeability and accelerates the excessive accumulation of ROS, which is associated with an imbalance among their assimilation and eradication [11,17]. The overproduction of ROS reacts with and devastates DNA, proteins, chlorophyll, and membrane phospholipids [30,36]. Regardless of their tremendously reactive and toxic nature, ROS are significant components of the signal transduction pathway, eliciting stress reactions, plant establishment, and productivity [37]. The initiation of plant oxidative impairment is a symbol of water deficit that is designated by a hyperaccumulation of H_2_O_2_, MDA, and PCG, and the dysfunction of cellular membranes associated with plant senescence [11,30]. In this investigation, water deficiency significantly increased H_2_O_2_ synthesis, leading to massive MDA and PCG accumulations, and induced EL% (Table 1). The current results concur with those of Hafez et al. [11] and Sadak et al. [38]. Additionally, the Si and/or Bc supplementation of drought-influenced borage plants relieved the oxidative injuries via lessening H_2_O_2_, MDA, and PCG assembly, therefore avoiding damage to cellular components [11,22,30]. As shown in Table 1, Si and/or Bc application mediates MDA reductions, and PCG accumulation is acceptable for the maintenance of cellular membrane structure and control of antioxidant capacity. Si and/or Bc possess(es) sufficient biochemical reactivates to elucidate their ability for acting as a facilitator of numerous physiological pathways and defense approaches [2,17]. The treated plant experience decreased cellular dehydration resulting from a chief desiccation-avoidance approach [11,22,30]—reserved crops delimited alongside ROS accumulation. Likewise, Si and/or Bc supplementation defends plants from oxidative damage by lessening superoxide anions and stimulating antioxidant enzymes, mainly SOD [2,17,22]. Such an influence would probably confirm the antioxidant approach of Si and/or Bc for overwhelming the higher levels of ROS elicited under water deficit.

Crops regularly produce numerous ROS through their photosynthetic and respiration processes in normal conditions. As a result, crops possess a distinct protection approach to eradicate ROS and sustain redox homeostasis, including the activation of antioxidant enzymes [2,17,22]. The control of antioxidant enzyme activities during water deficit is an adaptive method distinguished in several plants [2,15]. Antioxidant enzymes normally react with ROS and produce a harmless end-product. The conversion of superoxide to H_2_O_2_ is catalyzed by SOD, and subsequently by CAT and POD to H_2_O [2,17]. While CAT and POD activity continually declined under water deficit, their activity was higher in Bc- and/or Si-treated plants. In the most recent study, Si + Bc directly eradicated H_2_O_2_ and effected a superior activation of the antioxidant enzymes within water-deficient plants, thus permitting them to withstand water deficiency. The enhancement in the antioxidant enzyme activity could result from either an adaptive modification or silent gene transcription [39]. Accordingly, at the cellular level, Bc or Si may relieve oxidative injury under drought due to using metabolic pathways more effectively in scavenging ROS, leading to the enhanced integrity of cellular membranes. Consequently, Hasanuzzaman et al. [17] and Salim et al. [2] established that the supplementation of Bc and Si, respectively, activates SOD, CAT, and POD enzymes. 

Non-enzymatic antioxidant molecules are essential for maintaining cellular functioning, controlling antioxidant enzyme activities, and ultimately increasing stress tolerance. By neutralizing ROS and lowering cell membrane peroxidation, phenolic substances (phenol, flavonoid, and anthocyanin) under stress play a substantial role in the reduction of oxidative injury [8,40]. The current findings show that with drought, Si, or Bc application, soluble phenolic was recognized to accumulate in high concentrations. These results are comparable to those that have been observed previously under drought and Si or Bc treatment [8,15]. Since phenolic compounds have electron-donating agents, they comprise the majority of natural antioxidants [41]. Additionally, by tricking the hydroxyl radical, phenolic compounds can lower the production of ROS and boost the expression of the phenylamino lyase gene [42].

Numerous investigations have reported a rise in flavonoids related to water deficit and Si [8] or Bc [43] applications due to their ability to scavenge ROS. Flavonoids can be accepted as the second route of defense alongside oxidative injuries, which is motivated simply if the principal response of a rise in antioxidant enzyme activity is not sufficiently effective to eradicate extra ROS. In *Achillea pachycephala*, prolonged drought accelerates flavonoid assimilation and flavonoid biosynthesis gene expression. This, in turn, lessens overall oxidative stress, which is attributed to the accumulation of ROS [44]. Flavonoids are the prime extensive secondary metabolites that are chiefly restricted in vacuoles, detached from the chief ROS resources; thus, their significance as antioxidants is slight in entire cells, in spite of their outstanding antioxidant proficiency in vitro [45]. Additionally, flavonoids’ ability to function as potent antioxidants involves chelating Fe^2+^, which impedes ROS accumulation [46]. Although there has been a prime consideration of their antioxidant aptitude, there is encouraging evidence that flavonoids are not probable hydrogen-donating antioxidants. Nevertheless, they might exert modulatory schedules at protein and lipid kinase signaling pathways [47].

The over-accumulation of anthocyanins during drought suggests that they play various biological activities in such conditions [48]. Anthocyanins’ photoprotective properties and their roles in lowering ROS, defective excitation procedures, and reduced weakness to photoinhibition have all been formally established [49]. Anthocyanins help in preserving cell homeostasis by reducing H_2_O_2_ once the chloroplasts’ antioxidant enzymes have reached their maximum activity [50]. Anthocyanin has been reported to specifically act as an O_2_ and H_2_O_2_ scavenger [51]. Recently, an increase in anthocyanin assimilation was connected to a slight generation of H_2_O_2_ and an increase in antioxidant capacity, suggesting that anthocyanin has a signaling and an H_2_O_2_ eradication function [52].

The current findings showed that the ascorbate level in drought-affected borage plants was significantly elevated, and the addition of Si or Bc motivated further elevations that were corroborated by a decline in oxidative injuries. This finding agrees with Afshari et al. [8] for Si, Farouk and Al-Huqail [53] for Bc, and Farouk and Omar [15] for drought. Ascorbic acid is a critical antioxidant that contributes to the ascorbic-glutathione cycle within the ROS mitigation process, reducing oxidative damage [54]. Ascorbic acid is a particularly potent scavenger of ROS due to its capacity to donate electrons in the diverse enzymatic and non-enzymatic activities desired for the removal of O_2_^−^ and H_2_O_2_ [54]. It can act as a secondary antioxidant while reducing α-tocopherol and protecting carotenoids, or it can act as a primary antioxidant when ascorbate peroxidase is being enhanced [55]. An outstanding indicator of the reduction in ROS in drought-affected borage plants is the increase in ascorbic acid concentration caused by Bc and/or Si.

A plant species’ ability to withstand a water deficit depends on a number of metabolic pathways that sustain water availability, protect chloroplasts, and maintain ion homeostasis [15,56]. The principal pathways comprise the over-accumulation of organic solutes that regulate nutrients and water fluctuation, in addition to sustaining a proficient ROS-alleviating approach under environmental stress [57]. The present findings indicate a substantial rise in osmoprotectants, such as TAA, GlyBet, and TSC, with Si and/or Bc supplementation and watering regimes (Table 4). Under environmental stresses, osmotically adjustable solutes may help maintain the highest turgor and conformation of proteins and maintain cellular homeostasis [58]. Possible causes of the accumulation of TAA in the treated plant include protein starvation or protein assimilation inhibition [2].

Due to its benefits and efficacy as an osmolyte, GlyBet stands out as one of the most effective osmoprotectants. Formally, GlyBet protects stressed plants by stabilizing the quaternary structures of complex proteins and the oxygen-evolving photosystem II complex, as well as by sustaining OA aptitude [59]. Furthermore, GlyBet overproduction supports the maintenance of OA in the Si and/or Bc supplementation of well-watered or drought-affected borage plants. Treating borage plants with Si and/or Bc considerably boosts the GlyBet in the plant; this rise may have been caused by encouraging GlyBet assimilation [39].

Total soluble carbohydrates provided the chief components to plant cell Ψs, and they were also crucial for plant OA under drought [30,60] or Si and/or Bc application [2,30]. Soluble carbohydrates exhibit hormone-like activities and are able to act as major messengers in signaling. Soluble sugar-mediated signaling regulates growing processes; hence, it has been shown that their accumulation at later growth phases enhances plant establishment [61]. According to the accepted theory, TSC function as osmotica, protects certain macromolecules, aids in the stabilization of bio-membrane construction, scavenges ROS, and offers membrane protection [2,60]. Additionally, TSC accumulation has been associated with decreased cell metabolism and increased carbon use efficiency [62].

Maintaining cell turgor is the primary crop defense mechanism for lessening the injury of drought [22,63]. Drought, Si, or Bc treatment enhanced the accumulation of osmolytes, which may decrease the cell Ψs [22,64]. A reduction in Ψw successively helps to maintain cell Ψp by significantly dropping Ψs [22,57]. The present findings provide evidence that Si and/or Bc increased osmolyte buildup (Table 5), which may be linked to higher negative Ψs under normal or drought stress (Table 5). The decline in Ψw represents a decisive sign of water deficit maladies that impact water uptake into meristematic tissues [65]. The current results are consistent with earlier findings that indicate that Ψw, Ψp, and RWC declined considerably under stress conditions [22,66]. Maintaining water status and restored OA through coordination with greater osmoprotectant assimilation, preservation water, and sustaining Ψp, which is necessary for maintaining typical growth, may be the cause of the improved performance of crops under drought owing to Si and/or Bc supplementation [22,67]. Under drought, plants with Si and/or Bc application showed greater drops in Ψs than those that were not treated (Table 5). The minor Ψs rates could maintain leaf water status within drought conditions. Additionally, Si and/or Bc supplementation helped plants to maintain the high turgor pressure needed for stomatal opening, accelerating photosynthetic rate. As a result, the cell Ψs is reduced, which gradually improves the cell’s ability to maintain turgor pressure, which is necessary for enhancing water preservation, stomatal function, CO_2_ fixation, improving photosynthesis effectiveness, and raising drought tolerance [68].

Leaf RWC is an imperative attribute for evaluating the water status of plants and reflecting the enduring metabolic activities in plant tissues, and it may be utilized as a reliable indicator of drought tolerance. Recent findings indicated that RWC profoundly decreased within drought stress, which was reported previously for many crops [2,8,22]. The decline in RWC could be attributable to a lack of soil moisture or the weakness of root systems that are not capable of compensating for transpirational water [69]. Keeping a higher RWC denotes a recovered water status [70]. The current study proved that Bc and/or Si supplementation under normal or drought stress raised RWC compared to untreated plants. These findings were supported by several authors in various plant species [2,8] for Si; [22,71] for Bc. The encouraging impact of Bc on RWC may be connected to improving soil physio-chemical and biological properties [24]. Additionally, Si application considerably decreases transpiration rates [72] and increases light capture features [73]. Furthermore, Si treatment dramatically boosts the expression of aquaporin genes, which consecutively improves root water permeability and uptake under stress conditions [74]. Likewise, the modulation of aquaporin transport activity can occur at the post-transcriptional level due to an Si-induced reduction of oxidative stress and membrane injury [75]. In stressed crops, better adventitious root production, stomatal closure, or a decrease in transpiration may all contribute to an increase in water retention with Si and/or Bc.

Osmotic adjustment is recognized as an adaptation strategy against water deficits [68]. The present findings suggested that increased osmolyte accumulation and the subsequent stabilization of cellular membranes may be the causes of the increase in OA capacity within plants treated with Bc and/or Si under irrigation regimes [76,77]. In drought-affected plants, Si supplementation raised the expression of many aquaporin genes, while Si treatment increased the activity of key transcription factors involved in the drought response in rice [74]. However, tomato root hydraulic conductivity was improved, though in contrast, there was no effect of Si treatment on root OA or the expression of aquaporins in plants, which accumulate Si in much smaller concentrations [78]. Furthermore, Si was found to boost the amount of auxins, gibberellins, cytokinins, and abscisic acid in white lupin [79]. In soybean, Si enhanced gibberellins concentration while reducing the drought-induced rise in salicylic acid [80]. On the other hand, the role of OA in the ability of borage plants to withstand drought is still unclear. One potential elucidation for the current impacts could be the participation of Si and/or Bc in OA capacity and adventitious root growth, leading to increasing the capability of crops to maintain turgor within drought.

Chlorophyll performs a vital role in the biomass assembly process and photosynthesis pathways. Current findings prove that drought induced a sharp reduction in chlorophyll concentration, although Si and/or Bc supplementation increased chlorophyll concentration within normal and stressful conditions (Table 6). The relationship between chlorophyll deterioration and drought is fairly well understood [2,22]. This reduction may have been caused by the following: (1) Chloroplast ultrastructure disruption brought on by excessive ROS assembly [15]; (2) Decreased accessibility of Fe, which is necessary for the biosynthesis of the Chl precursors, i.e., protochlorophyllide and α-aminolevulinic acid ‘ALA’ [81]; (3) The variability of the pigment–protein complex coupled with chlorophyll oxidation [82]; (4) Destruction of the light-harvesting pigment–protein complexes [83]; (5) Elevation of Proto and Mg-Proto (Table 7), which prevent ALA assimilation [84]; and (6) Activation of proteolytic enzymes such as chlorophyllase [85]. The encouragement of Proto and Mg-Proto assimilation from glucose via a,a′-dipyridyl-induced porphyrin synthesis (a,a′DP, an iron chelator) occurs as the stage between glutamic and ALA, suppressed by heme assimilation in the chloroplasts, is prohibited by Fe sequestering mediators such as a,a′-DP [86]. The main Chl_a_ assimilate enzymes, HemA (glutamyl-tRNA reductase 1) and Chl H (magnesium chelatase H subunit), have persistently lower expression levels in response to PAO (PAO; up-regulated). Surprisingly, PAO has been strongly proven to be a Chl catabolic enzyme, and former investigations have shown that environmental stresses increase the expression of PAO in plants [87]. 

Due to enhanced Fe accessibility caused by Si and/or Bc supplementation, the transformation of Mg-Proto to Pchlide and subsequently Chl_a_ and Chl_b_ was accelerated under drought conditions. On the other hand, it has previously been proven that Si and/or Bc treatment increases chlorophyll accumulation in stress-affected plants: [2,77] for Si and [11,22] for Bc. It was noted that Si and/or Bc compensated the Mg-dechelatase and chlorophyllase activities that may have decreased the Chl deprivation. Additionally, Akhtar et al. [88,89] and Torabian et al. [90] discovered that Bc boosted chlorophyll concentration while decreasing ROS production and DPPH activity in leaf cells. Chl–protein complex (LHCP II-Chl) separation; Proto IX, Mg-Proto IX, Pchlide, Chl_a_ to Chl_b_ transformation; Chl dephytylation; Mg removal; and ring-opening to produce colorless primary fluorescent are thought to be the primary mechanisms of chlorophyll catabolism [91]. The present study found that a lack of water hastens the breakdown of Chl by increasing the concentration of porphyrin intermediates, which increases the risk of creating high-energy oxidizing elements that could speed up oxidative bleaching and then induce Chl deficiency. A previous study found that in drought-impacted plants, ROS assembly was noticeably raised but Chl was mostly decreased [15]; in contrast, abolishing impacts were presented with Si and/or Bc application, precisely ROS buildup, which was decreased, and the Chl was raised [53]. Furthermore, these findings suggest that Si and/or Bc may increase borage Chl by reducing the oxidative depletion of Chl.

Chlide is frequently regarded as a real intermediary of Chl biosynthesis or collapse that may occur during the production of Pheo_a_ as a Chl derivative [92]. According to Guyer et al. [93], pheophytin-specific phytolhydrolase (PPH) genes are found widely in several plants. This supports the hypothesis that chlorophyll degradation during droughts may frequently involve Pheo-specific dephytylation by PPHs [93]. After the formation of Chlide, one of the vinyl groups of the ring structure is reduced and a phytol chain is introduced to produce the decisive Chl_a_. A portion of Chl_a_ is transmuted to Chl_b_ in an oxygenase reaction. Controlling an assimilation cycle can happen at various altitudes. Through the use of expressional regulators, a variety of post-translational methods can be used to alter the activity of the enzymes, and the flux of early precursors into the route can be used to measure the enzymatic pathway. Following phototransformation, Chlide is converted into esterified pigments such as Chl_a_ and Chl_b_ [78]. 

Drought severely decreased leaves area per plant, which was confirmed by Ning et al. [77] and Ors et al. [94]. Drought reduces the turgor required for cell division and inhibits cell elongation [63,95]. Declined turgor pressure and a slow level of photosynthesis within a water deficit generally limit leaf expansion [96]. Additionally, a drought’s negative effects could be ascribed to a reduction in nutrient accessibility in the soil and a disruption in the nutritive state of plants connected with a lack of water supply [97,98]. In However, the application of Bc and/or Si significantly enhanced borage leaf area per plant under drought stress, which was previously observed: [77] for Si and [22] for Bc. Generally, Bc and/or Si may promote cell growth by loosening the cell wall, protecting the phospholipid bilayer, improving the fluidity of the membrane, and minimizing ROS damage: [77] for Si and [22] for Bc. Silicon can therefore control the formation of cell walls by encouraging tissue extensibility and accelerating cellular biochemical activities, increasing the stiffness of leaves [99]. Additionally, Bc treatment under stressful circumstances also boosted leaf area by preventing the synthesis of ABA [88].

As previously documented, plants suffer tremendously and produce less under stress [22,94,100]. This decrease is frequently attributed to the fact that drought has an adverse effect on all biochemical, morphological, and molecular characteristics, including (1) the prevention of water and photoassimilate translocation [22,77]; (2) the defeat of photosynthetic effectiveness and diminishing of ATP assimilation necessary for plant establishment [77]; (3) prevented ion uptake, which disturbs nutrient status and induces the production of ROS [97,98]; (4) decreased pollen grain viability [101]; and (5) disrupted assimilation movement to the growing fruits, which leads to total sterility [102]. Alternatively, under normal or drought stress, the supplementation of Si and/or Bc significantly increased borage plant productivity: [2] for Si and [22] for Bc. Numerous studies have shown that Bc and/or Si may enhance soil physical, chemical, and biological qualities and may enhance plant growth by delivering macro- and micronutrients [90,103]. Additionally, Bc can efficiently boost the soil’s capability for absorption and limit the leaching or volatilization of nutrients, which will facilitate nutrient absorption, slow their release, and improve crop yield [104]. Normally, Si and/or Bc increase(s) plant production, according to Sattar et al. [22], by boosting net photosynthetic rates, stomatal conductance, photosynthetic process, and reducing oxidative burst. Desoky et al. [30] also found that applying Si to drought-affected bean plants boosted yield due to an increase in photosynthetic rate, photosynthetic efficiency, chlorophyll assimilation, and a decrease in MDA assimilation.

## 4. Materials and Methods

### 4.1. Experimental Layout and Crop Husbandry

A split-plot in a completely randomized block design, with five replicates, was carried out in the Kalabsho region (Latitude: 31°12′52.27″ N; Longitude: 31°21′28.73″ E, 13 m elevation), Dakahlia Governorate, Egypt, during the 2018/2019 and 2019/2020 seasons. The main plots were designated for irrigation regimes, while the sub-plots were reserved for Bc and/or Si (Figure 2). The experimental sandy soil had an organic content of 1.53 and 1.59%, a pH of 7.91 and 7.86, and an EC of 2.9 and 2.89 dS m^−1^ in both years. The experimental district is a dry area with hot dry summers and low yearly precipitation in the winter. 

Before sowing, the experimental field was leveled and plowed. Fertilizers were added in accordance with the recommendation of the Egyptian Ministry of Agriculture and Land Reclamation. Farmyard manure of 11.90 m^3^ ha^−1^ was mixed with the surface layer during soil preparation, and Table 9 indicates the physical and chemical properties of farmyard manures used in the current experiment. Ammonium sulfate (33% N), calcium superphosphate (15.5% P_2_O_5_), and potassium sulfate (48% K_2_O) fertilizers (200, 200, 100 kg fed^−1^) were added. Nitrogen and potassium were used separately after 30 days from sowing and at full bloom, while phosphorous was added to the soil during field preparation.

Borage seeds were secured from Medicinal and Aromatic Plants Res. Dept., Agricultural Research Center, Egypt. Before sowing, following the recommendation of the Ministry of Agriculture and Land Reclamation, Egypt, borage seeds were inoculated with bioinoculants (*Azotobacter chroccoccum*, *Glomus mosseae*, and *Bacillus circulans*) that were delivered by the General Organization for Agriculture Equalization Fund, Egypt, at the rate of 4.76 kg ha^−1^. On 25th September in both seasons, the seeds were sown in hills (3–5 seeds per hill^−1^), 35 cm apart on the ridge, and then thinned to 1 plant per hill^−1^ 15 days after sowing. Plants were irrigated using a drip irrigation system (Figure 2A,B) for identical plant establishment via drippers of 4 L h^−1^ at 1 bar for four weeks before the irrigation regimes treatment. Irrigation regimes of 100% (well-watered, W), 75% (moderate drought, M), and 50% (severe drought, S) crop evapotranspiration were applied, i.e., 2336, 1752, and 1.168 m^3^ of water fed^−1^ (4200 m^2^) in the first season and 2283, 1712, and 1.141 m^3^ of water fed^−1^ in the second season. Results were calculated following the Penman–Montieth scheme [105] included in the CROPWAT 8″ model, which provides more constantly precise ETo assessments than other procedures of ETo.

The experimental plots received 9.52 tons ha^−1^ of Bc during the soil preparation. The physical and chemical characteristics of Bc can be observed in Table 10. Meanwhile, 300 mg L^−1^ of potassium silicate was sprayed at 45, 60, and 75 days after sowing, and water was used to spray the control plants. Tween 20 was added at a rate of 0.1% to the spraying solution as a surfactant to ensure the permanent spraying of solution into plant leaf tissues.

### 4.2. Data Recorded

Nine plants were randomly harvested 85 days from sowing to determine the morpho-physiological parameters for both seasons. The leaf area (cm^2^) was measured using a leaf area meter (model AM 300; ADC Bioscientific Ltd., Hoddesdon, Herts EN11 0NT, UK).

### 4.3. Oxidative Destruction Biomarkers

Leaf (1 g) was extracted with acetone (2 mL) to measure the concentration of H_2_O_2_ [82]. Leaf extract was combined with titanium reagent to produce a 2% titanium (Ti) solution (IV). Then, 0.2 mL of ammonia solution precipitated the Ti-H_2_O_2_ combination and non-reacted Ti. The precipitate was re-suspended in ice-cold acetone five times, drained, and then dissolved in 3 mL of 1 M H_2_SO_4_ before an assessment of absorbance at 410 nm. Lipid peroxidation was measured using the Shao et al. [55] protocol using the thiobarbituric acid (TBA) reactive compound aldehyde (MDA). The reaction with 5% TBA was carried out at 100 °C for 30 min and then cooled. After that, specific (532 nm) and non-specific (600 nm) optical density (OD) were recorded and represented as MDA nmol g^−1^ FW. The PCG, the final byproduct of protein oxidation, was assessed using a Levine et al. [106] scheme. We utilized a modified method of Tariq et al. [82] to determine the membrane permeability of borage plant tissues. Leaf segments were placed in a flask with 50 mL of distilled water for 4 h at laboratory temperature. The solution’s electrical conductivity (EC1) was then measured. The flasks were then placed in a boiling water bath for 30 min, and EC2 was recorded. Electrolyte leakage (EL) was deliberate through the subsequent equation EL% = (EC1/EC2) × 100.

### 4.4. Enzymatic Antioxidants Measurement

The Chrysargyris et al. [107] protocol was used to extract soluble protein and antioxidant enzymes from leaf tissues using 100 mM K-phosphate buffer (pH 7.0). Nitro-blue tetrazolium (NBT) photochemical reduction was utilized to measure SOD (EC 1.15.1.1) activity according to Chrysargyris et al. [107]. The OD was recorded at 560 nm and activity was expressed as unit mg^−1^ of protein (1 unit of SOD is the amount of enzyme anticipated to produce a 50% inhibition of the NBT photo-reduction).

The Chrysargyris et al. [107] method was used to evaluate the CAT (EC 1.11.1.6) activity as a decrease in the absorbance at 240 nm for 1 min next to the degradation of H_2_O_2_. A reaction mixture including an enzyme aliquot (1.5 mL), H_2_O_2_ (10 mM), and K-phosphate buffer (50 mM) was used. With the use of the H_2_O_2_ extinction coefficient (40 mM^−1^ cm^−1^), the activity of CAT (unit mg^−1^ protein) was calculated (one unit = 1 mM of H_2_O_2_ deterioration min^−1^). Alternatively, the Tarchoune et al. [108] modified technique was utilized to assess POD activity (EC 1.11.1.7). Through using the coefficient of extinction of 2.47 mM cm^−1^, the POD activity (unit mg^−1^ protein) was assessed, where 1 unit of POD = 1 mol of H_2_O_2_ degeneration min^−1^. The Bradford [109] technique was used to determine the amount of soluble protein in the extract. 

### 4.5. Antioxidant Solutes

The concentration of soluble phenol was measured via the Folin–Ciocalteu technique following the protocol of Sadasivam and Manickam [110]. The plant extract was combined with the Folin–Ciocalteu reagent and saturated sodium carbonate and agitated for 15 s before being allowed to sit for 30 min. The absorbance was subsequently measured using a T60 UV-Visible spectrophotometer (PG Instruments Ltd., Lutterworth, UK) at 760 nm. 

The flavonoid level (g quercetin equivalent g^−1^ FW) was estimated spectrophotometrically by the aluminum chloride (AlCl_3_) protocol [111]. Concisely, 2.8 mL of distilled water, 0.5 mL plant extract, 1.5 mL ethanol (95%), 0.1 mL AlCl_3_ (10%), and 0.1 mL potassium acetate (1M) were mixed together. The mixture was vortexed for 10 s before being settled for 30 min, and then OD was determined at 415 nm.

The anthocyanin (mg 100 g^−1^ FW) was extracted from borage leaves with acidified methanol, then the absorbance was noted at 530 nm. Through the use of a molar absorptive coefficient of 26,900, anthocyanin was estimated [112]. Using 2,6-dichlorophenol indophenols, ascorbic acid was assessed using the method described by Sadasivam and Manickam [110].

### 4.6. Organic Solutes Estimation

With the acid-ninhydrin reagent and glycine as a reference, TAA was assessed using the Dubey and Rani [113] methodology. The GlyBet concentration was deliberated following the Grieve and Grattan [114] scheme using an iodide reagent. According to the Zhang et al. [115] protocol, TSC was estimated spectrophotometrically using the anthrone reagent. Then, 100 µL of extract was immediately reacted with freshly made anthrone reagent (0.15 g anthrone + 0.1 L of H_2_SO_4_; 72%, *v*/*v*). The reacting combination was heated in a water bath for 10 min before cooling. A T60 UV-Visible spectrophotometer was used to measure absorbance at 625 nm.

### 4.7. Leaf Water Status Estimation

The procedure developed by Kaya et al. [116] was used to calculate leaf RWC. Leaf segments were weighed for fresh mass (FM) assessment. Subsequent leaves were floated in distilled water within a covered Petri plate to calculate the turgid mass (TM). Leaf samples were frequently weighed during the imbibition period after the water had been carefully wiped from the leaf surface using tissue paper. To obtain dry mass (DM), leaf samples were oven dried at 75 °C for 36 h. The following equation was utilized to determine LRWC: LRWC % = $\frac{FM-DM}{TM-DM}\times 100$

A Chardakoves scheme was used to evaluate the leaf water potential (Ψw) [95]. Meanwhile, osmotic potential (Ψs) was determined using the Van ’t Hoff equation: Ψs (MPa) = −miRT, where m, molality (moles per 1000 g); i, ionization constant; R, gas constant; and T, temperature (k). The difference between Ψw and Ψs was used to calculate the turgor potential (Ψp) [117]. The osmotic adjustment (OA) was assessed as the difference in Ψs between treated and un-treated plants [118].

### 4.8. Chlorophyll Concentration and Assimilation

A T60 UV-Visible spectrophotometer was used to assess the chlorophyll concentration of borage leaves once extracted with cold methanol under laboratory conditions for 48 h. Consistent with Harpaz-Saad et al. [119], chlorophyll and chlorophyllide (Chlide) concentrations were assessed. The concentrations of protoporphyrin (Proto), Mg-protoporphyrin (Mg-Proto), and protochlorophyllide (Pchlide) were calculated using the Sarropoulou et al. [120] protocol.

### 4.9. Yield Characteristics 

Plant seed yield −1 (g), seed index (g), fixed oil%, and plant fixed oil yield −1 were all measured after harvesting (180 days after sowing). Using a Soxhlet extractor for 6 h, fixed oil was extracted from air-dried seeds and then its percentage and oil yield per plant were considered.

### 4.10. Statistical Analysis

Following the data normality test, data were evaluated via a two-way analysis of variance (ANOVA) with the COSTATC statistical program (CoHort software, release 6.3.0.3, 2006; NC, USA). The Tukey’s Honestly Significant Difference (HSD) test was utilized for comparing means at *p* ≤ 0.05. The means and standard errors (SE) of five independent biological replications were used to present the data.

## 5. Conclusions

The current results would be very useful for permanent oil production within water-deficient environments. The improvement in antioxidant capacity and osmoprotectant accumulation positively reduced bio-membrane dysfunction and improved plant growth and productivity, which were linked to the motivation of water deficit tolerance in borage plants after Si and/or Bc treatment. Si and/or Bc are/is significantly less expensive than other materials and are/is appropriate for extensive use in agriculture production in the scenario of accelerating climate change. For forthcoming approaches, the mechanism of soil–plant relationships once Bc and/or Si are/is used must be considered.

## Figures and Tables

**Figure 1 plants-12-01605-f001:**
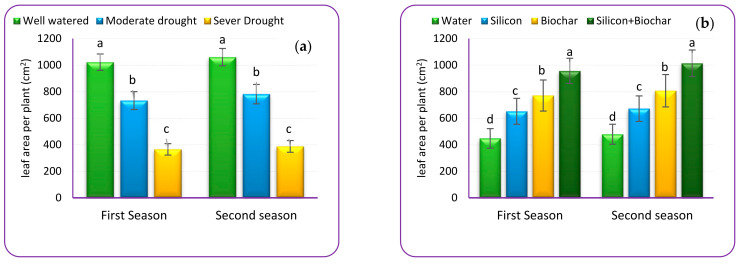
Effect of irrigation regimes (**a**), silicon, biochar (**b**) and their interaction (**c**) on leaf area (cm^2^) per borage plant throughout two experimental years. Mean values ± standard error are inside each column for each trial, with equivalent letters meaning not significantly different following Tukey test at *p* ≤ 0.05.

**Figure 2 plants-12-01605-f002:**
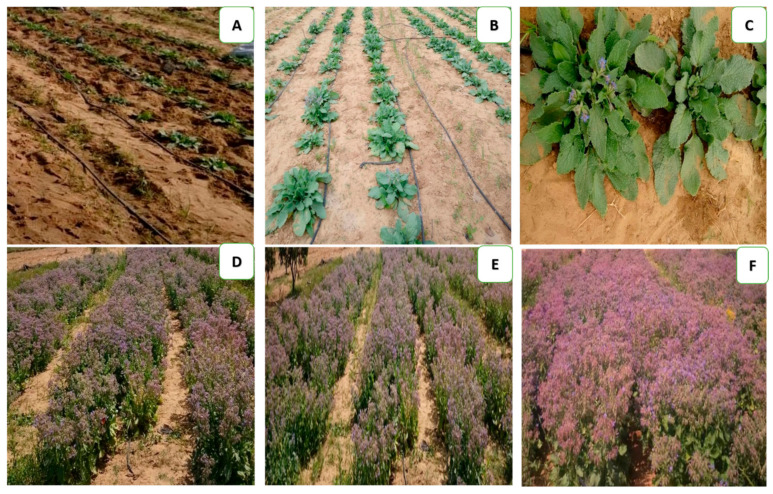
Layout of the experiment, (**A**) plants after thinning; (**B**) vegetative growth; (**C**) the beginning of flowering; (**D**–**F**) full flowering.

**Table 1 plants-12-01605-t001:** Effect of irrigation regimes, silicon, biochar, and their interaction on oxidative biomarkers of borage plants throughout two experimental years.

Treatment	Hydrogen Peroxide (micromole g^−1^ FW)		Malondialdehyde (nmol g^−1^ FW)		Protein Carbonyl Group (mM 100 g^−1^ FW)		Electrolyte Leakage (%)	
Season	First	Second	First	Second	First	Second	First	Second
Irrigation regimes								
Well water (W)	36.10 ± 2.63 ^c^	35.32 ± 2.64 ^c^	8.221 ± 0.39 ^c^	7.927 ± 0.38 ^c^	25.84 ± 0.92 ^b^	24.93 ± 0.95 ^b^	71.77 ± 1.23 ^b^	70.40 ± 1.30 ^c^
Moderate drought (M)	50.46 ± 2.26 ^b^	49.85 ± 2.25 ^b^	10.52 ± 0.38 ^b^	10.36 ± 0.40 ^b^	29.73 ± 0.30 ^b^	29.62 ± 0.32 ^b^	75.80 ± 1.35 ^b^	75.61 ± 0.80 ^b^
Severe drought (S)	71.76 ± 2.62 ^a^	69.28 ± 2.75 ^a^	13.95 ± 0.49 ^a^	13.23 ± 0.48 ^a^	37.73 ± 3.40 ^a^	37.82 ± 3.63 ^a^	80.69 ± 1.56 ^a^	80.75 ± 1.41 ^a^
ANOVA *p* values	***	***	***	***	***	***	***	***
Modulators								
Water (C)	62.74 ± 5.63 ^a^	61.72 ± 5.50 ^a^	12.68 ± 1.04 ^a^	12.45 ± 0.96 ^a^	38.85 ± 4.58 ^a^	39.28 ± 4.85 ^a^	79.21 ± 1.94	78.23 ± 2.47
Silicon (Si)	54.22 ± 5.06 ^b^	53.76 ± 5.17 ^b^	11.38 ± 0.83 ^b^	10.68 ± 0.65 ^b^	29.74 ± 0.82 ^b^	29.27 ± 0.89 ^b^	76.02 ± 2.11	75.55 ± 1.75
Biochar (Bc)	52.86 ± 5.08 ^b^	50.17 ± 4.29 ^b^	10.19 ± 0.64 ^c^	9.971 ± 0.67 ^b^	28.91 ± 0.98 ^b^	28.14 ± 0.95 ^b^	75.40 ± 1.87	75.20 ± 2.02
Si + Bc	41.29 ± 5.44 ^c^	40.29 ± 5.27 ^c^	9.344 ± 0.89 ^c^	8.931 ± 0.84 ^c^	26.90 ± 1.41 ^b^	26.49 ± 1.60 ^b^	73.72 ± 1.87	73.37 ± 1.44
ANOVA *p* values	***	***	***	***	***	***	ns	ns
Interaction effects								
W + C	43.18 ± 0.45 ^de^	42.46 ± 1.04 ^e–g^	9.252 ± 0.61 ^de^	9.188 ± 0.26 ^d–f^	29.02 ± 0.46 ^bc^	28.74 ± 0.55 ^bc^	74.49 ± 1.48	71.83 ± 5.11 ^b^
W + Si	40.62 ± 0.63 ^e^	40.31 ± 0.10 ^fg^	9.081 ± 0.09 ^de^	8.696 ± 0.05 ^ef^	27.22 ± 0.43 ^bc^	26.13 ± 0.25 ^cd^	71.81 ± 2.62	70.83 ± 1.64 ^b^
W + Bc	39.33 ± 1.16 ^e^	38.11 ± 0.30 ^g^	8.354 ± 0.04 ^e^	7.905 ± 0.04 ^f^	25.69 ± 1.34 ^cd^	24.65 ± 0.45 ^d^	70.71 ± 3.27	70.19 ± 2.39 ^b^
W + Si + Bc	21.29 ± 0.72 ^f^	20.40 ± 0.27 ^h^	6.196 ± 0.24 ^f^	5.918 ± 0.37 ^g^	21.43 ± 0.91 ^d^	20.22 ± 0.68 ^e^	70.07 ± 2.79	68.73 ± 0.21 ^b^
M + C	62.95 ± 1.31 ^c^	62.53 ± 1.36 ^c^	12.45 ± 0.24 ^c^	12.37 ± 0.47 ^b^	30.56 ± 0.73 ^bc^	30.51 ± 0.77 ^b^	78.70 ± 1.84	77.65 ± 1.04 ^ab^
M + Si	48.06 ± 0.68 ^d^	46.91 ± 0.64 ^e^	10.49 ± 0.15 ^d^	10.32 ± 0.55 ^cd^	29.86 ± 0.64 ^bc^	29.79 ± 0.54 ^b^	75.53 ± 4.77	75.14 ± 2.01 ^ab^
M + Bc	46.60 ± 0.38 ^d^	45.80 ± 0.57 ^e^	9.679 ± 0.61 ^de^	9.508 ± 0.31 ^de^	29.41 ± 0.58 ^bc^	29.13 ± 0.66 ^bc^	74.84 ± 1.71	74.74 ± 1.03 ^ab^
M + Si + Bc	44.24 ± 1.71 ^de^	44.18 ± 0.00 ^ef^	9.487 ± 0.27 ^de^	9.273 ± 0.20 ^d–f^	29.07 ± 0.34 ^bc^	29.06 ± 0.52 ^bc^	74.14 ± 2.27	74.93 ± 2.29 ^ab^
S + C	82.09 ± 0.11 ^a^	80.18 ± 2.13 ^a^	16.34 ± 0.03 ^a^	15.79 ± 0.07 ^a^	56.96 ± 2.15 ^a^	58.59 ± 1.11 ^a^	84.45 ± 3.86	85.21 ± 1.06 ^a^
S + Si	73.98 ± 0.00 ^b^	74.06 ± 0.58 ^b^	14.59 ± 0.33 ^b^	13.03 ± 0.29 ^b^	32.14 ± 1.19 ^b^	31.88 ± 0.91 ^b^	80.71 ± 2.00	80.69 ± 2.41 ^ab^
S + Bc	72.64 ± 1.52 ^b^	66.60 ± 1.86 ^c^	12.54 ± 0.09 ^c^	12.50 ± 0.00 ^b^	31.62 ± 0.71 ^b^	30.63 ± 0.71 ^b^	80.66 ± 1.87	80.66 ± 3.86 ^ab^
S + Si + Bc	58.35 ± 1.67 ^c^	56.29 ± 0.25 ^d^	12.35 ± 0.02 ^c^	11.60 ± 0.44 ^bc^	30.20 ± 0.65 ^bc^	30.20 ± 0.69 ^b^	76.95 ± 4.14	76.44 ± 1.76 ^ab^
ANOVA *p* values	***	***	***	***	***	***	ns	**

Levels of significance are represented by *** *p* < 0.001, ** *p* < 0.01, and ns means not significant. Mean values ± standard error are inside each column for each trial, with equivalent letters meaning not significantly different following Tukey test at *p* ≤ 0.052.

**Table 2 plants-12-01605-t002:** Effect of irrigation regimes, silicon, biochar and their interaction on some antioxidant activities (unit mg^−1^ protein) of borage plants throughout two experimental years.

Treatment	Superoxide Dismutase		Catalase		Peroxidase	
Season	First	Second	First	Second	First	Second
Irrigation regimes						
Well water (W)	61.24 ± 1.22 ^c^	59.66 ± 1.36 ^b^	56.20 ± 0.44 ^a^	55.50 ± 0.66 ^a^	35.38 ± 0.83 ^a^	34.39 ± 0.67 ^a^
Moderate drought (M)	67.05 ± 1.03 ^b^	66.56 ± 1.03 ^a^	50.53 ± 0.73 ^b^	49.83 ± 0.70 ^b^	26.92 ± 1.09 ^b^	26.37 ± 1.15 ^b^
Severe drought (S)	70.06 ± 0.99 ^a^	68.93 ± 0.90 ^a^	37.27 ± 3.10 ^c^	36.16 ± 2.92 ^c^	18.49 ± 1.33 ^c^	17.75 ± 1.28 ^c^
ANOVA *p* values	***	***	***	***	***	***
Modulators						
Water (C)	69.81 ± 1.59 ^a^	68.83 ± 1.52 ^a^	41.92 ± 4.57 ^b^	41.53 ± 4.36 ^b^	21.71 ± 3.08 ^c^	21.17 ± 3.12 ^c^
Silicon (Si)	66.51 ± 1.43 ^ab^	65.64 ± 1.34 ^ab^	45.23 ± 3.75 ^b^	44.37 ± 3.98 ^b^	27.29 ± 2.27 ^b^	26.57 ± 2.27 ^b^
Biochar (Bc)	65.10 ± 1.38 ^b^	64.08 ± 1.52 ^bc^	51.52 ± 1.81 ^a^	51.31 ± 1.28 ^a^	27.97 ± 2.34 ^b^	27.59 ± 2.32 ^ab^
Si + Bc	63.03 ± 1.92 ^b^	61.66 ± 2.18 ^c^	53.32 ± 1.27 ^a^	51.45 ± 2.11 ^a^	30.75 ± 2.21 ^a^	29.34 ± 2.09 ^a^
ANOVA *p* values	***	***	***	***	***	***
Interaction effects						
W + C	64.47 ± 1.76 ^b–d^	63.81 ± 1.86 ^a–c^	54.47 ± 0.38 ^bc^	53.39 ± 1.22 ^bc^	32.60 ± 1.54 ^b^	32.45 ± 0.77 ^ab^
W + Si	62.78 ± 1.75 ^b–d^	61.98 ± 1.73 ^b–d^	55.23 ± 0.47 ^a–c^	54.90 ± 0.10 ^ab^	34.81 ± 1.19 ^ab^	33.69 ± 1.34 ^ab^
W + Bc	61.18 ± 1.87 ^cd^	59.25 ± 1.92 ^cd^	57.17 ± 0.38 ^ab^	55.77 ± 1.24 ^ab^	35.56 ± 0.51 ^ab^	35.41 ± 0.35 ^a^
W + Si + Bc	56.51 ± 2.39 ^d^	53.61 ± 1.08 ^d^	57.92 ± 0.00 ^a^	57.92 ± 1.12 ^a^	38.57 ± 1.44 ^a^	36.00 ± 1.88 ^a^
M + C	71.23 ± 1.82 ^ab^	70.66 ± 1.54 ^a^	47.14 ± 0.75 ^fg^	46.60 ± 0.18 ^fg^	20.98 ± 0.69 ^fg^	20.12 ± 0.47 ^d^
M + Si	66.32 ± 1.61 ^a–c^	65.64 ± 1.75 ^a–c^	49.83 ± 0.49 ^d–f^	49.40 ± 0.77 ^d–f^	27.82 ± 0.26 ^de^	27.65 ± 0.21 ^c^
M + Bc	65.51 ± 1.72 ^a–d^	65.12 ± 1.67 ^a–c^	52.42 ± 0.85 ^cd^	50.59 ± 0.57 ^c–e^	28.84 ± 0.77 ^cd^	27.99 ± 0.33 ^c^
M + Si + Bc	65.12 ± 1.64 ^a–d^	64.84 ± 1.98 ^a–c^	52.75 ± 0.49 ^cd^	52.75 ± 0.18 ^b–d^	30.05 ± 0.50 ^cd^	29.73 ± 1.21 ^bc^
S + C	73.74 ± 1.06 ^a^	72.03 ± 1.60 ^a^	24.16 ± 0.10 ^i^	24.59 h ± 0.32 ^i^	11.54 ± 0.52 ^h^	10.94 ± 0.31 ^e^
S + Si	70.43 ± 2.06 ^ab^	69.29 ± 1.49 ^ab^	30.63 ± 1.14 ^h^	28.80 ± 0.56 ^h^	19.25 ± 0.25 ^g^	18.38 ± 0.71 ^d^
S + Bc	68.60 ± 1.61 ^a–c^	67.86 ± 1.39 ^ab^	44.98 ± 0.97 ^g^	47.57 ± 0.85 ^ef^	19.51 ± 0.54 f^g^	19.38 ± 0.29 ^d^
S + Si + Bc	67.46 ± 1.60 ^a–c^	66.55 ± 1.67 ^a–c^	49.29 ± 0.70 ^ef^	43.68 ± 0.49 ^g^	23.65 ± 0.62 ^ef^	22.31 ± 0.65 ^d^
ANOVA *p* values	***	***	***	***	***	***

Levels of significance are represented by *** *p* < 0.001. Mean values ± standard error are inside each column for each trial, with equivalent letters meaning not significantly different following Tukey test at *p* ≤ 0.05.

**Table 3 plants-12-01605-t003:** Effect of irrigation regimes, silicon, biochar and their interaction on antioxidant solute concentrations of borage plants throughout two experimental years.

Treatment	Phenol (mg Gallic Acid g^−1^ FW)		Flavonoid (µg Quercetin g^−1^ FW)		Anthocyanin (mg 100 g^−1^ FW)		Ascorbic Acid (mg g^−1^ FW)	
Season	First	Second	First	Second	First	Second	First	Second
Irrigation regimes								
Well water (W)	11.78 ± 0.60 ^c^	12.08 ± 0.58 ^c^	2.213 ± 0.14 ^c^	2.283 ± 0.12 ^c^	3.331 ± 0.30 ^c^	3.469 ± 0.29 ^c^	11.38 ± 1.02 ^b^	12.36 ± 1.13 ^b^
Moderate drought (M)	17.11 ± 0.97 ^b^	18.26 ± 0.87 ^b^	3.222 ± 0.18 ^b^	3.251 ± 0.18 ^b^	4.708 ± 0.17 ^b^	5.019 ± 0.08 ^b^	16.61 ± 0.19 ^a^	16.65 ± 0.20 ^a^
Severe drought (S)	24.60 ± 0.89 ^a^	24.82 ± 0.92 ^a^	3.930 ± 0.07 ^a^	4.097 ± 0.10 ^a^	5.634 ± 0.16 ^a^	5.714 ± 0.18 ^a^	17.36 ± 0.15 ^a^	17.53 ± 0.15 ^a^
ANOVA *p* values	***	***	***	***	***	***	***	***
Modulators								
Water (C)	13.82 ± 1.61 ^c^	14.04 ± 1.59 ^c^	2.491 ± 0.31 ^c^	2.618 ± 0.29 ^c^	3.523 ± 0.41 ^d^	3.891 ± 0.45 ^d^	12.97 ± 1.63 ^c^	13.04 ± 1.62 ^b^
Silicon (Si)	17.35 ± 1.98 ^b^	18.97 ± 1.87 ^b^	3.041 ± 0.25 ^b^	3.064 ± 0.26 ^b^	4.397 ± 0.37 ^c^	4.515 ± 0.33 ^c^	15.00 ± 0.94 ^b^	15.11 ± 0.91 ^ab^
Biochar (Bc)	19.68 ± 2.04 ^a^	19.71 ± 1.86 ^b^	3.433 ± 0.25 ^a^	3.430 ± 0.25 ^a^	4.942 ± 0.35 ^b^	5.008 ± 0.34 ^b^	15.44 ± 0.93 ^ab^	16.73 ± 0.39 ^a^
Si + Bc	20.48 ± 1.90 ^a^	20.83 ± 2.06 ^a^	3.522 ± 0.21 ^a^	3.730 ± 0.27 ^a^	5.368 ± 0.21 ^a^	5.522 ± 0.23 ^a^	17.06 ± 0.30 ^a^	17.17 ± 0.30 ^a^
ANOVA *p* values	***	***	***	***	***	***	ns	ns
Interaction effects								
W + C	8.577 ± 0.23 ^f^	8.959 ± 0.51 ^f^	1.391 ± 0.04 ^e^	1.624 ± 0.08 ^e^	1.971 ± 0.03 ^g^	2.101 ± 0.20 ^g^	6.466 ± 0.29 ^g^	6.600 ± 0.11 ^i^
W + Si	12.27 ± 0.25 ^e^	12.32 ± 0.08 ^e^	2.356 ± 0.08 ^d^	2.390 ± 0.07 ^d^	2.954 ± 0.02 ^f^	3.247 ± 0.08 ^f^	11.26 ± 0.13 ^f^	11.46 ± 0.06 ^h^
W + Bc	12.34 ± 0.22 ^e^	13.08 ± 0.19 ^de^	2.436 ± 0.01 ^cd^	2.428 ± 0.03 ^d^	3.636 ± 0.05 ^e^	3.747 ± 0.04 ^e^	11.73 ± 0.06 ^f^	15.26 ± 0.17 ^g^
W + Si + Bc	13.93 ± 0.26 ^d^	13.96 ± 0.27 ^d^	2.669 ± 0.02 ^c^	2.690 ± 0.03 ^d^	4.763 ± 0.02 ^d^	4.782 ± 0.02 ^cd^	16.06 ± 0.26 ^de^	16.13 ± 0.13 ^ef^
M + C	13.22 ± 0.08 ^de^	13.28 ± 0.08 ^de^	2.546 ± 0.02 ^cd^	2.550 ± 0.01 ^d^	3.769 ± 0.09 ^e^	4.716 ± 0.02 ^d^	15.80 ± 0.20 ^e^	15.66 ± 0.17 ^fg^
M + Si	14.63 ± 0.31 ^d^	19.32 ± 0.19 ^c^	2.719 ± 0.08 ^c^	2.732 ± 0.10 ^d^	4.795 ± 0.01 ^d^	4.811 ± 0.02 ^cd^	16.40 ± 0.20 ^c–e^	16.53 ± 0.06 ^de^
M + Bc	20.24 ± 0.11 ^c^	20.13 ± 0.08 ^c^	3.794 ± 0.12 ^ab^	3.785 ± 0.03 ^bc^	5.091 ± 0.04 ^b–d^	5.159 ± 0.09 ^bc^	17.06 ± 0.06 ^a–d^	17.13 ± 0.24 ^b–d^
M + Si + Bc	20.37 ± 0.24 ^c^	20.33 ± 0.28 ^c^	3.828 ± 0.05 ^ab^	3.680 ± 0.04 ^c^	5.175 ± 0.08 ^bc^	5.387 ± 0.10 ^b^	17.20 ± 0.34 ^a–c^	17.26 ± 0.17 ^bc^
S + C	19.68 ± 0.40 ^c^	19.88 ± 0.22 ^c^	3.536 ± 0.05 ^b^	2.398 ± 0.04 ^b^	4.830 ± 0.03 ^cd^	4.855 ± 0.03 ^cd^	16.66 ± 0.24 ^b–e^	16.86 ± 0.06 ^bc^
S + Si	25.15 ± 0.34 ^b^	25.28 ± 0.31 ^b^	4.048 ± 0.01 ^a^	4.069 ± 0.17 ^b^	5.441 ± 0.12 ^b^	5.488 ± 0.06 ^b^	17.33 ± 0.17 ^a–c^	17.33 ± 0.06 ^bc^
S + Bc	26.45 ± 0.41 ^ab^	25.93 ± 0.27 ^b^	4.069 ± 0.02 ^a^	4.077 ± 0.02 ^b^	6.099 ± 0.10 ^a^	6.118 ± 0.09 ^a^	17.53 ± 0.06 ^ab^	17.80 ± 0.11 ^ab^
S + Si + Bc	27.12 ± 0.33 ^a^	28.20 ± 0.11 ^a^	4.069 ± 0.06 ^a^	4.564 ± 0.08 ^a^	6.167 ± 0.12 ^a^	6.397 ± 0.01 ^a^	17.93 ± 0.17 ^a^	18.13 ± 0.17 ^a^
ANOVA *p* values	***	***	***	***	***	***	ns	**

Levels of significance are represented by *** *p* < 0.001, ** *p* < 0.01, and ns means not significant. Mean values ± standard error are inside each column for each trial, with equivalent letters meaning not significantly different following Tukey test at *p* ≤ 0.05.

**Table 4 plants-12-01605-t004:** Effect of irrigation regimes, silicon, biochar and their interaction on some organic solute concentrations of borage plants throughout two experimental years.

Treatment	Free Amino Acids (mg g^−1^ DW)		Glycinebetaine (mg g^−1^ DW)		Soluble Carbohydrates (µg g^−1^ DW)	
Season	First	Second	First	Second	First	Second
Irrigation regimes						
Well water (W)	20.57 ± 0.65 ^c^	20.20 ± 0.63 ^c^	12.63 ± 0.21 ^b^	12.12 ± 0.35 ^c^	94.07 ± 5.05 ^c^	91.42 ± 3.97 ^c^
Moderate drought (M)	30.51 ± 0.59 ^b^	28.73 ± 0.94 ^b^	14.35 ± 0.20 ^b^	14.07 ± 0.23 ^b^	155.9 ± 5.52 ^b^	149.9 ± 5.06 ^b^
Severe drought (S)	42.20 ± 1.42 ^a^	40.97 ± 1.37 ^a^	18.73 ± 1.11 ^a^	17.93 ± 0.76 ^a^	198.2 ± 5.19 ^a^	189.3 ± 2.31 ^a^
ANOVA *p* values	***	***	***	***	***	***
Modulators						
Water (C)	27.75 ± 3.08 ^c^	26.07 ± 3.05 ^c^	13.11 ± 0.56 ^b^	13.14 ± 0.78 ^c^	131.0 ± 16.3 ^c^	128.1 ± 15.2 ^c^
Silicon (Si)	30.06 ± 2.75 ^bc^	29.14 ± 2.71 ^b^	14.47 ± 0.70 ^b^	14.20 ± 0.63 ^bc^	142.9 ± 14.3 ^bc^	139.0 ± 14.6 ^b^
Biochar (Bc)	31.56 ± 2.77 ^b^	30.49 ± 2.64 ^b^	14.92 ± 0.74 ^b^	14.76 ± 0.75 ^b^	152.3 ± 14.8 ^b^	145.7 ± 14.0 ^b^
Si + Bc	35.01 ± 3.99 ^a^	34.18 ± 3.77 ^a^	17.74 ± 1.80 ^a^	16.73 ± 1.37 ^a^	171.4 ± 16.3 ^a^	161.2 ± 13.3 ^a^
ANOVA *p* values	***	***	***	***	***	***
Interaction effects						
W + C	17.17 ± 0.25 ^f^	16.92 ± 0.24 ^e^	11.96 ± 0.37 ^f^	10.33 ± 0.20 ^g^	74.72 ± 3.37 ^h^	76.47 ± 2.49 ^h^
W + Si	20.47 ± 0.16 ^ef^	19.96 ± 0.25 ^de^	12.33 ± 0.18 ^ef^	12.26 ± 0.31 ^fg^	91.45 ± 11.38 ^gh^	85.04 ± 4.28 ^gh^
W + Bc	21.86 ± 0.15 ^e^	21.53 ± 0.14 ^d^	12.86 ± 0.37 ^d–f^	12.56 ± 0.31 ^f^	97.39 ± 3.43 ^gh^	93.51 ± 1.64 ^g^
W + Si + Bc	22.78 ± 0.41 ^e^	22.39 ± 0.12 ^d^	13.36 ± 0.33 ^d–f^	13.33 ± 0.29 ^ef^	112.73 ± 6.31 ^fg^	110.64 ± 0.80 ^f^
M + C	27.66 ± 0.49 ^d^	23.75 ± 0.27 ^d^	13.73 ± 0.20 ^c–f^	13.40 ± 0.20 ^ef^	134.02 ± 10.75 ^ef^	126.69 ± 4.93 ^e^
M + Si	30.13 ± 0.17 ^cd^	28.88 ± 0.29 ^c^	14.10 ± 0.00 ^c–f^	13.86 ± 0.46 ^d–f^	150.38 ± 1.31 ^de^	146.71 ± 1.34 ^d^
M + Bc	31.78 ± 0.12 ^c^	30.16 ± 0.40 ^c^	14.16 ± 0.12 ^c–f^	14.16 ± 0.29 ^d–f^	161.96 ± 6.10 ^c–e^	154.14 ± 1.32 ^d^
M + Si + Bc	32.47 ± 0.84 ^c^	32.13 ± 0.32 ^c^	15.40 ± 0.35 ^b–e^	14.86 ± 0.55 ^c–e^	177.24 ± 4.09 ^b–d^	172.19 ± 2.14 ^c^
S + C	38.42 ± 1.13 ^b^	37.53 ± 1.41 ^b^	15.73 ± 0.39 ^b–d^	15.70 ± 0.35 ^b–d^	184.27 ± 8.57 ^bc^	181.34 ± 1.92 ^bc^
S + Si	39.57 ± 0.32 ^b^	38.56 ± 1.27 ^b^	17.00 ± 0.58 ^bc^	16.46 ± 0.24 ^bc^	186.89 ± 3.27 ^bc^	185.47 ± 0.63 ^b^
S + Bc	41.05 ± 0.39 ^b^	39.77 ± 0.53 ^b^	17.73 ± 0.26 ^b^	17.56 ± 0.23 ^b^	197.67 ± 2.94 ^ab^	189.63 ± 2.07 ^ab^
S + Si + Bc	49.77 ± 1.82 ^a^	48.01 ± 1.79 ^a^	24.46 ± 2.01 ^a^	22.00 ± 0.91 ^a^	224.26 ± 0.09 ^a^	200.78 ± 2.08 ^a^
ANOVA *p* values	***	***	***	***	***	***

Levels of significance are represented by *** *p* < 0.001. Mean values ± standard error are inside each column for each trial, with equivalent letters meaning not significantly different following Tukey test at *p* ≤ 0.05.

**Table 5 plants-12-01605-t005:** Effect of irrigation regimes, silicon, biochar and their interaction on water status of borage plants throughout two experimental years.

Treatment	Relative Water Content		Water Potential (-MPa)		Osmotic Potential (-MPa)		Turger Potential (MPa)		Osmotic Adjustment	
Season	First	Second	First	Second	First	Second	First	Second	First	Second
Irrigation regimes										
Well water (W)	72.22 ± 1.14 ^a^	71.58 ± 1.10 ^a^	0.953 ± 0.049 ^a^	0.919 ± 0.050 ^a^	1.100 ± 0.009 ^a^	1.081 ± 0.010 ^a^	0.149 ± 0.046 ^b^	0.161 ± 0.043 ^b^	0.018 ±0.004 ^c^	0.039 ± 0.007 ^c^
Moderate drought (M)	67.80 ± 1.10 ^b^	67.34 ± 0.80 ^b^	1.304 ± 0.030 ^b^	1.255 ± 0.029 ^b^	1.537 ± 0.011 ^b^	1.496 ± 0.016 ^b^	0.233 ± 0.020 ^a^	0.240 ± 0.018 ^ab^	0.455 ± 0.015 ^b^	0.451 ± 0.019 ^b^
Severe drought (S)	60.91 ± 1.24 ^c^	59.63 ± 1.38 ^c^	1.543 ± 0.059 ^c^	1.535 ± 0.017 ^c^	1.88 ± 0.015 ^c^	1.817 ± 0.024 ^c^	0.273 ± 0.045 ^a^	0.306 ± 0.013 ^a^	0.735 ± 0.019 ^a^	0.792 ± 0.026 ^a^
ANOVA *p* values	***	***	***	***	***	***	***	***	***	***
Modulators										
Water (C)	64.32 ± 2.25 ^b^	64.07 ± 2.07 ^b^	1.133 ± 0.118 ^a^	1.081 ± 0.118 ^a^	1.467 ± 0.107 ^a^	1.379 ± 0.093 ^a^	0.334 ± 0.017 ^a^	0.298 ± 0.026 ^a^	0.301 ± 0.107 ^c^	0.337 ± 0.114 ^c^
Silicon (Si)	67.95 ± 1.19 ^ab^	66.47 ± 1.93 ^ab^	1.317 ± 0.073 ^b^	1.276 ± 0.075 ^b^	1.511 ± 0.113 ^b^	1.500 ± 0.112 ^bc^	0.194 ± 0.042 ^b^	0.224 ± 0.039 ^ab^	0.429 ± 0.114 ^ab^	0.455 ± 0.107 ^ab^
Biochar (Bc)	65.18 ± 2.31 ^b^	64.59 ± 2.26 ^b^	1.234 ± 0.083 ^ab^	1.223 ± 0.083 ^b^	1.489 ± 0.110 ^ab^	1.467 ± 0.107 ^b^	0.254 ± 0.029 ^ab^	0.243 ± 0.026 ^ab^	0.406 ± 0.110 ^b^	0.425 ± 0.112 ^b^
Si + Bc	70.45 ± 1.87 ^a^	69.60 ± 1.91 ^a^	1.389 ± 0.126 ^c^	1.366 ± 0.080 ^c^	1.556 ± 0.120 ^c^	1.511 ± 0.113 ^c^	0.173 ± 0.024 ^c^	0.178 ± 0.034 ^b^	0.473 ± 0.121 ^a^	0.492 ± 0.093 ^a^
ANOVA *p* values	***	***	***	***	***	***	***	***	***	***
Interaction effects										
W + C	69.90 ± 3.27 ^ab^	69.89 ± 1.71 ^ab^	0.689 ± 0.010 ^a^	0.647 ± 0.012 ^a^	1.082 ± 0.017 ^a^	1.042 ± 0.016 ^a^	0.402 ± 0.026 ^a^	0.395 ± 0.010 ^a^	0 ± 0.00 ^j^	0 ± 0.00 ^i^
W + Si	71.73 ± 0.75 ^ab^	71.65 ± 1.63 ^a^	1.064 ± 0.011 ^c^	1.020 ± 0.018 ^c^	1.103 ± 0.018 ^a^	1.098 ± 0.018 ^a^	0.039 ± 0.028 ^f^	0.078 ± 0.035 ^ef^	0.021 ± 0.001 ^h^	0.056 ± 0.003 ^h^
W + Bc	70.60 ± 0.03 ^ab^	69.90 ± 3.26 ^ab^	0.950 ± 0.003 ^b^	0.937 ± 0.012 ^b^	1.092 ± 0.017 ^a^	1.082 ± 0.017 ^a^	0.142 ± 0.014 ^de^	0.145 ± 0.020 ^de^	0.010 ± 0.010 ^hi^	0.040 ± 0.002 ^i^
W + Si + Bc	76.64 ± 1.92 ^a^	74.87 ± 1.48 ^a^	1.109 ± 0.003 ^c^	1.074 ± 0.020 ^cd^	1.124 ± 0.018 ^a^	1.103 ± 0.018 ^a^	0.015 ± 0.021 ^f^	0.029 ± 0.004 ^f^	0.042 ± 0.001 ^g^	0.061 ± 0.003 ^h^
M + C	66.32 ± 1.83 ^bc^	65.78 ± 1.17 ^a–c^	1.213 ± 0.009 ^d^	1.131 ± 0.012 ^d^	1.499 ± 0.010 ^b^	1.410 ± 0.007 ^b^	0.286 ± 0.002 ^bc^	0.279 ± 0.007 ^bc^	0.417 ± 0.027 ^f^	0.368 ± 0.023 ^g^
M + Si	67.55 ± 1.32 ^ab^	67.55 ± 1.31 ^ab^	1.317 ± 0.015 ^e^	1.267 ± 0.010 ^e^	1.543 ± 0.012 ^bc^	1.531 ± 0.011 ^c^	0.226 ± 0.003 ^cd^	0.264 ± 0.021 ^bc^	0.461 ± 0.029 ^de^	0.489 ± 0.027 f
M + Bc	67.46 ± 3.95 ^ab^	66.92 ± 1.71 ^a–c^	1.227 ± 0.012 ^d^	1.220 ± 0.010 ^e^	1.520 ± 0.010 ^bc^	1.499 ± 0.010 ^c^	0.293 ± 0.002 ^bc^	0.279 ± 0.002 ^bc^	0.438 ± 0.027 ^e^	0.457 ± 0.026 ^f^
M + Si + Bc	69.88 ± 1.66 ^ab^	69.11 ± 2.34 ^ab^	1.459 ± 0.012 ^f^	1.402 ± 0.008 ^f^	1.587 ± 0.014 ^c^	1.543 ± 0.012 ^c^	0.128 ± 0.023 ^e^	0.141 ± 0.020 ^de^	0.505 ± 0.030 ^c^	0.501 ± 0.028 ^e^
S + C	56.74 ± 0.87 ^c^	56.53 ± 0.61 ^c^	1.497 ± 0.007 ^fg^	1.464 ± 0.008 ^fg^	1.569 ± 0.011 ^d^	1.687 ± 0.010 ^d^	0.072 ± 0.012 ^ef^	0.223 ± 0.014 ^cd^	0.487 ± 0.027 ^d^	0.645 ± 0.026 ^d^
S + Si	64.59 ± 1.40 ^bc^	60.23 ± 2.61 ^bc^	1.571 ± 0.020 ^h^	1.541 ± 0.010 ^hi^	1.888 ± 0.011 ^de^	1.871 ± 0.010 ^e^	0.317 ± 0.022 ^ab^	0.330 ± 0.009 ^ab^	0.806 ± 0.027 ^b^	0.829 ± 0.025 ^b^
S + Bc	57.48 ± 1.39 ^c^	56.95 ± 1.48 ^c^	1.526 ± 0.009 ^gh^	1.513 ± 0.009 ^gh^	1.854 ± 0.011 ^d^	1.821 ± 0.011 ^e^	0.328 ± 0.011 ^ab^	0.308 ± 0.013 ^bc^	0.772 ± 0.027 ^bc^	0.779 ± 0.026 ^c^
S + Si + Bc	64.85 ± 0.72 ^bc^	64.81 ± 3.29 ^a–c^	1.581 ± 0.014 ^i^	1.623 ± 0.009 ^i^	1.957 ± 0.011 ^e^	1.988 ± 0.011 ^e^	0.376 ± 0.005 ^a^	0.365 ± 0.011 ^bc^	0.875 ± 0.026 ^a^	0.946 ± 0.025 ^a^
ANOVA *p* values	***	***	***	***	***	***	***	***	***	***

Levels of significance are represented by *** *p* < 0.001. Mean values ± standard error are inside each column for each trial, with equivalent letters meaning not significantly different following Tukey test at *p* ≤ 0.05.

**Table 6 plants-12-01605-t006:** Effect of irrigation regimes, silicon, biochar and their interaction on chlorophyll_a_ and chlprophyll_b_ concentration (mg g^−1^ FW) as well as the ratio of chlorophyll_a_/chlorophyllide_a_ and chlorophyll_b_/chlorophyllide_b_ of borage plants throughout two experimental years.

Treatment	Chlorophyll_a_		Chlorophyll_b_		Chlorophyll_a_/Chlorophyllide_a_		Chlorophyll_b_/Chlorophyllide_b_	
Season	First	Second	First	Second	First	Second	First	Second
Irrigation regimes								
Well water (W)	1.699 ± 0.016 ^a^	1.679 ± 0.012 ^a^	0.997 ± 0.028 ^a^	0.989 ± 0.020 ^a^	0.740 ± 0.025 ^a^	0.703 ± 0.018 ^a^	0.417 ± 0.013 ^a^	0.401 ± 0.007 ^a^
Moderate drought (M)	1.631 ± 0.002 ^b^	1.623 ± 0.004 ^a^	0.948 ± 0.008 ^a^	0.928 ± 0.015 ^b^	0.520 ± 0.035 ^b^	0.578 ± 0.040 ^ab^	0.309 ± 0.011 ^b^	0.325 ± 0.014 ^b^
Severe drought (S)	1.479 ± 0.029 ^c^	1.349 ± 0.083 ^b^	0.827 ± 0.010 ^b^	0.768 ± 0.033 ^c^	0.550 ± 0.056 ^b^	0.545 ± 0.072 ^b^	0.339 ± 0.023 ^b^	0.341 ± 0.032 ^b^
ANOVA *p* values	***	**	**	***	**	*	***	**
Modulators								
Water (C)	1.526 ± 0.052 ^b^	1.370 ± 0.124 ^b^	0.893 ± 0.031 ^b^	0.774 ± 0.050 ^b^	0.576 ± 0.055 ^a^	0.489 ± 0.060 ^b^	0.332 ± 0.018 ^a^	0.297 ± 0.026 ^b^
Silicon (Si)	1.599 ± 0.024 ^a^	1.585 ± 0.027 ^a^	0.944 ± 0.035 ^ab^	0.931 ± 0.026 ^a^	0.654 ± 0.076 ^a^	0.501 ± 0.063 ^b^	0.384 ± 0.035 ^a^	0.311 ± 0.022 ^b^
Biochar (Bc)	1.632 ± 0.032 ^a^	1.608 ± 0.024 ^a^	0.966 ± 0.036 ^a^	0.954 ± 0.034 ^a^	0.596 ± 0.059 ^a^	0.731 ± 0.027 ^a^	0.353 ± 0.026 ^a^	0.410 ± 0.016 ^a^
Si + Bc	1.654 ± 0.027 ^a^	1.638 ± 0.031 ^a^	0.894 ± 0.017 ^ab^	0.919 ± 0.026 ^a^	0.589 ± 0.034 ^a^	0.714 ± 0.025 ^a^	0.351 ± 0.013 ^a^	0.404 ± 0.010 ^a^
ANOVA *p* values	**	**	**	**	ns	***	ns	***
Interaction effects								
W + C	1.639 ± 0.004 ^b^	1.639 ± 0.005 ^c^	0.977 ± 0.018 ^bc^	0.895 ± 0.012 ^cd^	0.735 ± 0.043 ^a–c^	0.651 ± 0.004 ^bc^	0.389 ± 0.008 ^a–c^	0.383 ± 0.006 ^bc^
W + Si	1.663 ± 0.008 ^b^	1.650 ± 0.005 ^bc^	1.057 ± 0.036 ^ab^	1.008 ± 0.024 ^ab^	0.816 ± 0.034 ^a^	0.749 ± 0.040 ^ab^	0.455 ± 0.035 ^a^	0.398 ± 0.012 ^ab^
W + Bc	1.741 ± 0.027 ^a^	1.680 ± 0.003 ^b^	1.093 ± 0.024 ^a^	1.075 ± 0.005 ^a^	0.775 ± 0.003 ^ab^	0.767 ± 0.001 ^ab^	0.450 ± 0.01 ^ab^	0.441 ± 0.003 ^a^
W + Si + Bc	1.753 ± 0.020 ^a^	1.746 ± 0.008 ^a^	0.862 ± 0.017 ^d–f^	0.977 ± 0.019 ^bc^	0.636 ± 0.042 ^c^	0.644 ± 0.002 ^bc^	0.374 ± 0.011 ^a–c^	0.382 ± 0.004 ^bc^
M + C	1.623 ± 0.004 ^bc^	1.598 ± 0.002 ^d^	0.926 ± 0.020 ^c–e^	0.846 ± 0.019 ^d^	0.623 ± 0.006 ^c^	0.563 ± 0.011 ^c^	0.340 ± 0.009 ^cd^	0.302 ± 0.005 ^de^
M + Si	1.630 ± 0.004 ^bc^	1.629 ± 0.003 ^cd^	0.948 ± 0.015 ^cd^	0.950 ± 0.012 ^bc^	0.357 ± 0.006 ^d^	0.379 ± 0.004 ^d^	0.250 ± 0.006 ^e^	0.263 ± 0.005 ^e^
M + Bc	1.632 ± 0.004 ^b^	1.631 ± 0.004 ^cd^	0.956 ± 0.017 ^c^	0.951 ± 0.018 ^bc^	0.641 ± 0.006 ^c^	0.637 ± 0.005 ^bc^	0.339 ± 0.017 ^cd^	0.346 ± 0.010 ^cd^
M + Si + Bc	1.638 ± 0.002 ^b^	1.6340.004 ^cd^	0.960 ± 0.010 ^c^	0.963 ± 0.016 ^bc^	0.460 ± 0.003 ^d^	0.733 ± 0.043 ^ab^	0.308 ± 0.005 ^c–e^	0.388 ± 0.008 ^bc^
S + C	1.315 ± 0.009 ^f^	1.477 ± 0.005 ^f^	0.775 ± 0.004 ^f^	0.581 ± 0.030 ^e^	0.369 ± 0.005 ^d^	0.253 ± 0.013 ^d^	0.267 ± 0.009 ^de^	0.206 ± 0.014 ^f^
S + Si	1.505 ± 0.008 ^e^	0.872 ± 0.012 ^g^	0.827 ± 0.005 ^f^	0.835 ± 0.004 ^d^	0.790 ± 0.045 ^ab^	0.373 ± 0.005 ^d^	0.448 ± 0.015 ^ab^	0.271 ± 0.009 ^e^
S + Bc	1.523 ± 0.005 ^de^	1.513 ± 0.007 ^ef^	0.847 ± 0.005 ^ef^	0.837 ± 0.005 ^d^	0.372 ± 0.006 ^d^	0.789 ± 0.046 ^a^	0.271 ± 0.010 ^de^	0.444 ± 0.016 ^a^
S + Si + Bc	1.571 ± 0.006 ^cd^	1.533 ± 0.012 ^e^	0.861 ± 0.008 ^d–f^	0.818 ± 0.007 ^d^	0.671 ± 0.006 ^bc^	0.766 ± 0.044 ^ab^	0.371 ± 0.024 ^bc^	0.441 ± 0.012 ^a^
ANOVA *p* values	***	***	***	***	***	***	***	***

Levels of significance are represented by * *p* < 0.05, ** *p* < 0.01, *** *p* < 0.001 and ns means not significant. Mean values ± standard error are inside each column for each trial, with equivalent letters meaning not significantly different following Tukey test at *p* ≤ 0.05.

**Table 7 plants-12-01605-t007:** Effect of irrigation regimes, silicon, biochar and their interaction on porphyrin intermediates of borage plants throughout two experimental years.

Treatment	Protoporphyrin		Mg Protoporphyrin		Protochlorophyllide	
Season	First	Second	First	Second	First	Second
Irrigation regimes						
Well water (W)	0.576 ± 0.005 ^b^	0.517 ± 0.010 ^b^	0.383 ± 0.004 ^c^	0.359 ± 0.012 ^b^	0.727 ± 0.030 ^c^	0.645 ± 0.057 ^b^
Moderate drought (M)	0.567 ± 0.004 ^a^	0.579 ± 0.003 ^a^	0.415 ± 0.002 ^b^	0.411 ± 0.003 ^a^	0.876 ± 0.019 ^b^	0.867 ± 0.011 ^a^
Severe drought (S)	0.558 ± 0.007 ^a^	0.556 ± 0.002 ^a^	0.428 ± 0.004 ^a^	0.426 ± 0.003 ^a^	1.048 ± 0.063 ^a^	0.936 ± 0.025 ^a^
ANOVA *p* values	***	***	***	***	***	***
Modulators						
Water (C)	0.581 ± 0.009 ^a^	0.574 ± 0.002 ^a^	0.419 ± 0.007 ^a^	0.417 ± 0.006 ^a^	1.040 ± 0.094 ^a^	0.912 ± 0.040 ^a^
Silicon (Si)	0.557 ± 0.005 ^b^	0.561 ± 0.003 ^a^	0.414 ± 0.007 ^a^	0.410 ± 0.006 ^a^	0.889 ± 0.029 ^b^	0.862 ± 0.020 ^a^
Biochar (Bc)	0.556 ± 0.006 ^ab^	0.548 ± 0.010 ^ab^	0.408 ± 0.006 ^ab^	0.398 ± 0.010 ^a^	0.834 ± 0.031 ^b^	0.797 ± 0.041 ^ab^
Si + Bc	0.527 ± 0.008 ^b^	0.517 ± 0.013 ^b^	0.394 ± 0.009 ^b^	0.370 ± 0.019 ^b^	0.772 ± 0.052 ^b^	0.693 ± 0.088 ^b^
ANOVA *p* values	**	***	***	***	***	***
Interaction effects						
W + C	0.559 ± 0.001 ^cd^	0.557 ± 0.004 ^b–d^	0.398 ± 0.004 ^b–d^	0.397 ± 0.008 ^cd^	0.817 ± 0.015 ^cd^	0.809 ± 0.018 ^b^
W + Si	0.528 ± 0.002 ^e^	0.538 ± 0.001 ^e^	0.390 ± 0.002 ^c–e^	0.387 ± 0.002 ^de^	0.804 ± 0.018 ^cd^	0.794 ± 0.022 ^b^
W + Bc	0.548 ± 0.001 ^d^	0.505 ± 0.002 ^f^	0.386 ± 0.001 ^de^	0.360 ± 0.003 ^e^	0.716 ± 0.024 ^de^	0.637 ± 0.018 ^c^
W + Si + Bc	0.509 ± 0.002 ^f^	0.466 ± 0.002 ^g^	0.358 ± 0.003 ^e^	0.292 ± 0.006 ^f^	0.569 ± 0.012 ^e^	0.342 ± 0.017 ^d^
M + C	0.591 ± 0.002 ^cd^	0.607 ± 0.001 ^ab^	0.423 ± 0.002 ^a–c^	0.421 ± 0.002 ^a–c^	0.901 ± 0.074 ^bc^	0.898 ± 0.021 ^ab^
M + Si	0.585 ± 0.002 ^c^	0.583 ± 0.002 ^b^	0.417 ± 0.002 ^a–d^	0.413 ± 0.005 ^a–d^	0.881 ± 0.036 ^bc^	0.875 ± 0.019 ^ab^
M + Bc	0.570 ± 0.004 ^b^	0.579 ± 0.002 ^a^	0.413 ± 0.001 ^a–d^	0.410 ± 0.008 ^a–d^	0.873 ± 0.021 ^b–d^	0.850 ± 0.021 ^b^
M + Si + Bc	0.561 ± 0.001 ^cd^	0.547 ± 0.001 ^c–e^	0.405 ± 0.005 ^a–d^	0.401 ± 0.001 ^b–d^	0.849 ± 0.020 ^b–d^	0.843 ± 0.022 ^b^
S + C	0.610 ± 0.006 ^a^	0.559 ± 0.003 ^bc^	0.435 ± 0.017 ^a^	0.434 ± 0.011 ^a^	1.401 ± 0.029 ^a^	1.029 ± 0.082 ^a^
S + Si	0.559 ± 0.003 ^cd^	0.564 ± 0.003 ^de^	0.434 ± 0.011 ^a^	0.429 ± 0.005 ^ab^	0.982 ± 0.033 ^b^	0.917 ± 0.014 ^ab^
S + Bc	0.552 ± 0.002 ^cd^	0.561 ± 0.002 ^e^	0.426 ± 0.004 ^ab^	0.425 ± 0.002 ^a–c^	0.911 ± 0.013 ^bc^	0.903 ± 0.019 ^ab^
S + Si + Bc	0.511 ± 0.003 ^cd^	0.540 ± 0.002 ^c–e^	0.419 ± 0.001 ^a–d^	0.417 ± 0.002 ^a–d^	0.897 ± 0.024 ^bc^	0.895 ± 0.023 ^ab^
ANOVA *p* values	***	***	***	***	***	***

Levels of significance are represented by *** *p* < 0.001, and ** *p* < 0.01. Mean values ± standard error are inside each column for each trial, with equivalent letters meaning not significantly different following Tukey test at *p* ≤ 0.05.

**Table 8 plants-12-01605-t008:** Effect of irrigation regimes, silicon, biochar and their interaction on borage plant yield throughout two experimental years.

Treatment	Seed Yield (g Plant^−1^)		Seed Index (g)		Oil %		Oil Yield (g Plant^−1^)	
Season	First	Second	First	Second	First	Second	First	Second
Irrigation regimes								
Well water (W)	9.924 ± 0.231 ^a^	11.30 ± 0.382 ^a^	15.89 ± 0.477 ^a^	15.46 ± 0.489 ^a^	29.70 ± 0.73 ^a^	30.42 ± 0.74 ^a^	2.966 ± 0.138 ^a^	3.469 ± 0.197 ^a^
Moderate drought (M)	9.200 ± 0.218 ^b^	9.936 ± 0.308 ^b^	14.28 ± 0.382 ^b^	13.56 ± 0.459 ^b^	26.80 ± 0.72 ^b^	27.49 ± 0.72 ^b^	2.482 ± 0.126 ^b^	2.756 ± 0.162 ^b^
Severe drought (S)	7.260 ± 0.238 ^c^	7.962 ± 0.255 ^c^	10.35 ± 0.574 ^c^	11.63 ± 0.252 ^c^	23.23 ± 0.49 ^c^	23.83 ± 0.51 ^c^	1.699 ± 0.091 ^c^	1.911 ± 0.101 ^c^
ANOVA *p* values	***	***	***	***	***	***	***	***
Modulators								
Water (C)	7.818 ± 0.364 ^d^	8.437 ± 0.408 ^d^	11.55 ± 0.860 ^d^	11.91 ± 0.399 ^d^	23.86 ± 0.72 ^d^	24.22 ± 0.75 ^d^	1.887 ± 0.139 ^d^	2.087 ± 0.159 ^d^
Silicon (Si)	8.446 ± 0.454 ^c^	9.197 ± 0.478 ^c^	12.54 ± 0.882 ^c^	13.18 ± 0.599 ^c^	25.71 ± 1.02 ^c^	26.36 ± 1.02 ^c^	2.207 ± 0.199 ^c^	2.463 ± 0.217 ^c^
Biochar (BI)	9.072 ± 0.414 ^b^	10.17 ± 0.476 ^b^	14.32 ± 0.891 ^b^	13.99 ± 0.668 ^b^	27.26 ± 1.02 ^b^	27.98 ± 1.02 ^b^	2.504 ± 0.200 ^b^	2.884 ± 0.242 ^b^
Si + BI	9.841 ± 0.361 ^a^	11.13 ± 0.607 ^a^	15.62 ± 0.671 ^a^	15.37 ± 0.724 ^a^	29.49 ± 1.07 ^a^	30.20 ± 1.09 ^a^	2.933 ± 0.205 ^a^	3.415 ± 0.296 ^a^
ANOVA *p* values	***	***	***	***	***	***	***	***
Interaction effects								
W + C	8.756 ± 0.044 ^e^	9.626 ± 0.053 ^e^	13.71 ± 0.017 ^e^	13.10 ± 0.017 ^e^	25.89 ± 0.02 ^e^	26.58 ± 0.03 ^e^	2.267 ± 0.011 ^e^	2.558 ± 0.017 ^ef^
W + Si	9.816 ± 0.048 ^c^	10.65 ± 0.084 ^d^	15.13 ± 0.034 ^d^	15.45 ± 0.014 ^d^	29.44 ± 0.01 ^c^	30.05 ± 0.01 ^c^	2.890 ± 0.013 ^c^	3.203 ± 0.026 ^d^
W + BI	10.28 ± 0.053 ^b^	11.96 ± 0.104 ^b^	16.92 ± 0.020 ^b^	16.61 ± 0.026 ^b^	31.14 ± 0.03 ^b^	31.87 ± 0.03 ^b^	3.203 ± 0.015 ^b^	3.813 ± 0.036 ^b^
W + Si + BI	10.83 ± 0.034 ^a^	12.96 ± 0.078 ^a^	17.80 ± 0.031 ^a^	17.41 ± 0.017 ^a^	32.36 ± 0.04 ^a^	33.18 ± 0.05 ^a^	3.507 ± 0.015 ^a^	4.303 ± 0.031 ^a^
M + C	8.316 ± 0.023 ^f^	8.813 ± 0.031 ^f^	12.80 ± 0.032 ^g^	12.24 ± 0.026 ^h^	24.65 ± 0.11 ^g^	25.24 ± 0.11 ^g^	2.050 ± 0.015 ^g^	2.224 ± 0.017 ^g^
M + Si	8.796 ± 0.012 ^e^	9.533 ± 0.063 ^e^	13.31 ± 0.034 ^f^	12.71 ± 0.017 ^f^	25.32 ± 0.05 ^f^	26.10 ± 0.05 ^f^	2.227 ± 0.001 ^e^	2.488 ± 0.021 f
M + BI	9.440 ± 0.015 ^d^	9.813 ± 0.062 ^e^	15.13 ± 0.023 ^d^	13.18 ± 0.025 ^e^	26.46 ± 0.03 ^d^	27.14 ± 0.04 ^d^	2.497 ± 0.005 ^d^	2.663 ± 0.021 ^e^
M + Si + BI	10.24 ± 0.063 ^b^	11.58 ± 0.070 ^c^	15.89 ± 0.023 ^c^	16.14 ± 0.014 ^c^	30.80 ± 0.07 ^b^	31.50 ± 0.08 ^b^	3.156 ± 0.026 ^b^	3.650 ± 0.031 ^c^
S + C	6.383 ± 0.044 ^i^	6.873 ± 0.035 ^h^	8.156 ± 0.057 ^j^	10.39 ± 0.029 ^j^	21.05 ± 0.18 ^j^	21.52 ± 0.18 i	1.344 ± 0.020 ^j^	1.479 ± 0.018 i
S + Si	6.726 ± 0.028 ^h^	7.403 ± 0.066 ^g^	9.170 ± 0.052 ^i^	11.38 ± 0.032 i	22.37 ± 0.03 ^i^	22.94 ± 0.04 ^h^	1.505 ± 0.008 ^i^	1.698 ± 0.018 ^h^
S + BI	7.490 ± 0.055 ^g^	8.733 ± 0.036 ^f^	10.90 ± 0.026 ^h^	12.20 ± 0.015 ^h^	24.20 ± 0.05 ^h^	24.92 ± 0.05 ^g^	1.812 ± 0.017 ^h^	2.176 ± 0.013 ^g^
S + Si + BI	8.440 ± 0.020 ^f^	8.840 ± 0.037 ^f^	13.18 ± 0.017 ^f^	12.57 ± 0.026 ^g^	25.30 ± 0.04 f	25.93 ± 0.04 ^f^	2.135 ± 0.008 ^f^	2.292 ± 0.013 ^g^
ANOVA *p* values	***	***	***	***	***	***	***	***

Levels of significance are represented by *** *p* < 0.001. Mean values ± standard error are inside each column for each trial, with equivalent letters meaning not significantly different following Tukey test at *p* ≤ 0.05.

**Table 9 plants-12-01605-t009:** Physical and chemical properties of Farmyard manure used in the current experiment.

Properties	pH	EC (dsm^−1^)	Organic Carbon (%)	Total N (%)	Total P (%)	Total K (%)	CEC (Cmol kg^−1^)	Moisture (%)
Values in the first year	8.790	3.160	58%	1.140	0.590	0.630	69.73	25.97%
Values in the second year	8.681	3.151	63%	1.145	0.589	0.632	68.92	26.11

pH: potential of hydrogen; EC: Electrical conductivity; N: Nitrogen; P: Phosphorus; K: Potassium; CEC: Cation exchange capacity.

**Table 10 plants-12-01605-t010:** Physical and chemical properties of biochar used in the current experiment.

Seasons	Properties						
	Organic Matter (%)	Total Carbon (C%)	Total Nitrogen (N%)	C/N Ratio	Total Phosphate (%)	Total Potassium (%)	pH (1:5 *w*/*v*)
First season	65.40	38.01	1.77	1:21.22	1.03	0.99	8.16
Second season	67.98	39.50	1.81	1:21.57	0.91	0.91	8.14

pH: potential of hydrogen.

## Data Availability

Not applicable.

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
