# Peer review of "Potential Role of Biochar and Silicon in Improving Physio-Biochemical and Yield Characteristics of Borage Plants under Different Irrigation Regimes"

_plants, 2023, doi:10.3390/plants12081605_

Round 1

Reviewer 1 Report

The manuscript by S. Farouk and co-autors titled “Potential role of biochar and silicon in improving physio-biochemical and yield characteristics of borage plants under different irrigation regimes” had the purpose to study the impact of Si and Bc on yield, water status, chlorophyll and solute accumulation, and antioxidant capacity of borage plants, assessment whether Si and Bc are extra effective in lessening drought in borage, and elucidate potential strategies by which Si and Bc lighten drought injuries in borage plants. Manuscript is interesting, the research results are practical and can be used in horticultural production, especially in conditions of water deficit. However, the publication requires the introduction of necessary corrections and supplementations indicated below for the Authors' consideration. Methods of the experiment are incompletely described. Please describe in detail the composition and application of the bioinoculants (you described the species of bacteria, but didn’t describe the species of mycorrhizal fungi). In the method chapter, please provide the chemical composition of the biochar used (necessarily) and the farmyard manure. Please state clearly how many plants constituted one replication and how many replications there were in the experiment. Please add photos documenting your experiments. Explain in Fig. 1 the notations a, b and c. The manuscript discussion is successful and the references are adequate. The experimental results are sufficient to justify the conclusions. In my opinion the paper is acceptable for publication in Plants after necessary revision.

Author Response

Dear prospective reviewer
We appreciate the time and effort you have dedicated to providing insightful feedback on ways to strengthen our manuscript. We have incorporated changes that reflect the detailed suggestions you have graciously provided. We think that the manuscript has been greatly improved by these revisions and hope that our edits and the responses we provide below satisfactorily address all the issues and concerns you have noted. The necessary corrections have been done with fast-track. To facilitate your evaluation, the following is a point-by-point response to the questions and comments

1- The manuscript by S. Farouk and co-autors titled “Potential role of biochar and silicon in improving physio-biochemical and yield characteristics of borage plants under different irrigation regimes” had the purpose to study the impact of Si and Bc on yield, water status, chlorophyll, and solute accumulation, and antioxidant capacity of borage plants, assessment whether Si and Bc are extra effective in lessening drought in borage, and elucidate potential strategies by which Si and Bc lighten drought injuries in borage plants. Manuscript is interesting, the research results are practical and can be used in horticultural production, especially in conditions of water deficit. However, the publication requires the introduction of necessary corrections and supplementations indicated below for the Authors' consideration. 

Thank you so much for your energetic encouragement, and we will do our best to increase the value of the current manuscript alongside your comments.

2- Methods of the experiment are incompletely described. Please describe in detail the composition and application of the bioinoculants (you described the species of bacteria, but didn’t describe the species of mycorrhizal fungi).

Thank you so much for your valuable comments, we checked this section as per your suggestion. As indicated in the revised manuscript we add the scientific name of mycorrhizal fungi as Glomus mosseae that was added following the recommendation of the Ministry of Agriculture and and Land Reclamation recommendation

3- In the method chapter, please provide the chemical composition of the biochar used (necessarily) and the farmyard manure.

Thank you so much for your valuable comments, we fully agree with your comment we add the physical and chemical analysis of Biochar (Table 10) and farmyard manure (Table 9) in the revised manuscript with the material and method section

4- Please state clearly how many plants constituted one replication and how many replications there were in the experiment.

Thank you for your recommendation, as indicated in the revised manuscript we used 9 plants from each treatment that is divided into 3 replicate (each replicate contain 3 plants)

5- Please add photos documenting your experiments.

Thank you as indicated in revised manuscript we add Figure 2 that contained some plant life cycle

6- Explain in Fig. 1 the notations a, b and c. 

Thank you for your suggestion we nominate it

7- The manuscript discussion is successful and the references are adequate.

Thank you so much for your energetic encouragement

8- The experimental results are sufficient to justify the conclusions.

Thank you so much for your energetic encouragement

 9- In my opinion the paper is acceptable for publication in Plants after necessary revision.

Thank you so much for your energetic encouragement, hoping to find the new revised version of our manuscript is good and to be more readable

Once more, thank you for giving us the opportunity to strengthen our manuscript with your valuable comments and queries. We have worked hard to incorporate your feedback and hope that these revisions persuade you to accept our submission.

Thank you in advance for your time and attention.

Sincerely

Prof. Dr. Saad Farouk

Agric. Botany Dept.

Fac. of Agric.

Mansoura Univ.

Egypt

Reviewer 2 Report

General comments

The article is interesting and innovative, however there are some points to improve.

Specific comments:

Abstract

Line 18. Fed-1 This is not a unit of the international system of units. The same applies to lines 577, 579 and 585.

Materials and Methods:

4.1. Experimental layout and Crop husbandry:

Line 569. The coordinates of the centre of the experimental plot of borages should be given.

Line 577. The characteristics of the manure used, especially its moisture content, should be specified.

Discussion:

The discussion is long but it lacks a few sentences with more real practical application of the experimentation carried out.

References:

It seems that there is quite a large number of references uses. Difficult to judge whether those are all relevant (and necessary) to the paper. Please take a critical look.

Author Response

Dear prospective reviewer
We appreciate the time and effort you have dedicated to providing insightful feedback on ways to strengthen our manuscript. We have incorporated changes that reflect the detailed suggestions you have graciously provided. We think that the manuscript has been greatly improved by these revisions and hope that our edits and the responses we provide below satisfactorily address all the issues and concerns you have noted. The necessary corrections have been done with fast-track. To facilitate your evaluation, the following is a point-by-point response to the questions and comments

1- General comments

The article is interesting and innovative, however there are some points to improve.

Thank you so much for your energetic encouragement, and we will do our best to increase the value of the current manuscript alongside your comments.

2- Specific comments:

Abstract

Line 18. Fed-1 This is not a unit of the international system of units. The same applies to lines 577, 579 and 585.

Thank you so much for your valuable comments, we fully agree with your comment and changed all units within the manuscript to the international system of units, thank you again

3- Materials and Methods:

4.1. Experimental layout and Crop husbandry:

Line 569. The coordinates of the centre of the experimental plot of borages should be given.

Thank you so much for your valuable comments, as indicated in the revised manuscript within the Material and Method section we add the coordinates of the experimental field as

“”A split-plot in a completely randomized block design, with five replicates, was carried out in the Kalabsho region (Latitude: 31° 12' 52.27" N; Longitude: 31° 21' 28.73" E, 13m Elevation), Dakhlia governorate, Egypt……””

4- Line 577. The characteristics of the manure used, especially its moisture content, should be specified.

Thank you so much for your valuable comments, we fully agree with your comment we add the physical and chemical analysis of farmyard manure (Table 9) in the revised manuscript with the material and method section

5- Discussion:

The discussion is long but it lacks a few sentences with more real practical applications of the experimentation carried out.

Thank you so much for your valuable comments, we checked the discussion section, thank you again

6- References:

It seems that there is quite a large number of references uses. Difficult to judge whether those are all relevant (and necessary) to the paper. Please take a critical look.

Thank you so much for your valuable comments, we checked all references used and find that all references were relevant to the manuscript, thank you again

Once more, thank you for giving us the opportunity to strengthen our manuscript with your valuable comments and queries. We have worked hard to incorporate your feedback and hope that these revisions persuade you to accept our submission.

Thank you in advance for your time and attention.

Sincerely

Prof. Dr. Saad Farouk

Agric. Botany Dept.

Fac. of Agric.

Mansoura Univ.

Egypt

Round 2

Reviewer 2 Report

Changes have been made